



# Multi-decadal mass balance series of three Kyrgyz glaciers inferred from transient snowline observations

Martina Barandun[1], Matthias Huss[1,2], Etienne Berthier[3], Andreas Kääb[4], Erlan Azisov[5], Tobias Bolch[6], Ryskul Usubaliev[5], and Martin Hoelzle[1]

[1]Department of Geosciences, University of Fribourg, Fribourg, Switzerland
[2]Laboratory of Hydraulics, Hydrology and Glaciology (VAW), ETH Zurich, Zurich, Switzerland
[3]CNRS, LEGOS, University of Toulouse, Toulouse, France
[4]Department of Geoscience, University of Oslo, Norway
[5]Central Asian Institute of Applied Geosciences (CAIAG), Bishkek, Kyrgyzstan
[6]Department of Geography, University of Zurich, Switzerland

*Correspondence to:* Martina Barandun (martina.barandun@unifr.ch)

**Abstract.**

Glacier mass balance observations in the Tien Shan and Pamir mountains are sparse and often discontinuous. Nevertheless, glaciers are one of the most important components of the high-mountain cryosphere in the region; they strongly influence water availability in the arid, continental and intensely populated downstream areas. This study provides reliable and continuous mass
balance series for selected glaciers located in the Tien Shan and Pamir-Alay. A combination of three independent methods was used to reconstruct for the past two decades the mass balance of the three benchmark glaciers, Abramov, Golubin and No. 354. By applying different approaches, it was possible to compensate for the limitations and shortcomings of each individual method. This study proposes the use of transient snowline observations throughout the melting season obtained from satellite imagery and terrestrial automatic cameras. By combining modelling with remotely acquired information on summer snow
depletion, it was possible to infer glacier mass changes for unmeasured years. Multi-annual mass changes based on high-accuracy digital elevation models and in situ glaciological surveys were used to validate the results for the investigated glaciers. Substantial mass loss was confirmed for the three studied glaciers by all three methods, ranging from $-0.30 \pm 0.19$ m w.e. a$^{-1}$ to $-0.41 \pm 0.33$ m w.e. a$^{-1}$ over the 2004-2016 period. Our results indicate that integration of snowline observations into mass balance modelling significantly narrows the uncertainty ranges of the estimates, and hence highlights the potential of the
methodology for application to inaccessible glaciers at larger scales for which no direct measurements are available.

## 1  Introduction

Glaciers are important components of the hydrological cycle in Central Asia. In this arid continental region, the intensely populated and irrigated downstream areas strongly depend on a supply of water from the cryosphere such as glaciers and snow (Kaser et al., 2010; Schaner et al., 2012; Duethmann et al., 2014; Chen et al., 2016; Huss et al., 2017). The uncertainty of water
availability in the context of a changing climate creates a major potential for political tension and builds a complex set of future threats, affecting different sectors such as water management, energy production and irrigation (Varis, 2014; Munia et al.,



2016; Pritchard, 2017). Climate change poses a manifold challenge for the Central Asian population and will influence natural hazards and threaten future economies and the livelihood of coming generations (IPCC Climate Change, 2013). For this reason, continuous and high-quality data for the different components of the hydrological cycle acquired within established regional and national cryospheric and hydrologic climate services are key for providing accurate predictions which enable sustainable

adaptation. As stated by the World Meteorological Organization (GCOS, 2016), large gaps currently exist in the global climate observation system. This refers in equal measure to such remote and inaccessible areas, as the Pamir and the Tien Shan, where there is a lack of data crucially needed to plan and enhance future development (Sorg et al., 2012; Unger-Shayesteh et al., 2013). Improved temporal and spatial representation of glacier monitoring is thus essential, due to the paramount signification of glaciers in the high-mountain cryosphere.

During the Soviet era, in the 1950s and 1960s, an extensive system of cryospheric monitoring was launched in the Tien Shan and Pamir region. Most programmes stopped abruptly with the breakdown of the USSR in the mid-1990s. Monitoring activities were maintained only on Tuyuksu Glacier, Kazakhstan, and Urumqi Glacier (No. 1), China. In recent years, different initiatives have aimed at the re-establishment of glacier monitoring in Central Asia (Hoelzle et al., 2017). Mass balance series are now available for the following glaciers: Abramov (Pamir-Alay), Batysh Sook, Sary-Tor, Karabatkak and No. 354 (Central

Tien Shan), for Urumchi No. 1 (Eastern Tien Shan) and for Golubin and Tuyuksu (Northern Tien Shan) (Fig. 1) (WGMS, 2013). However, a significant gap in the data from the mid-1990s to around 2010 hinder the interpretation of long-term trends in glacier behaviour in this region.

Different studies derived continuous mass balance series for selected glaciers based on modelling (e.g.,,, Fujita et al., 2011; Barandun et al., 2015; Kronenberg et al., 2016; Liu and Liu, 2016; Kenzhebaev et al., 2017) and estimated mass balance at a

regional scale in the Pamir-Alay and Tien Shan (Farinotti et al., 2015). Modelled mass balance series have a good temporal resolution; however, they are not observation-based and thus strongly depend on model calibration and the quality of the input variables. Several studies use remote sensing techniques to fill the gaps in glacier monitoring and to generate insights into region-wide mass changes covering entire High Mountain Asia within the past two decades (e.g., Gardner et al., 2013; Gardelle et al., 2013; Kääb et al., 2015; Brun et al., 2017; Wang et al., 2017). Furthermore, other authors focused on selected

regions of the Central and Northern Tien Shan (e.g., Aizen et al., 2007; Pieczonka et al., 2013; Bolch, 2015; Pieczonka and Bolch, 2015; Goerlich et al., 2017) deriving glacier-specific geodetic mass balances. These studies often cover large areas, but temporal resolution is typically limited to five years or longer periods. Thus, they fail to capture the interannual or even seasonal signals.

The snowline is recognized as a valuable proxy for glacier mass balance (LaChapelle, 1962; Lliboutry, 1965; Braithwaite,

1984; Kulkarni, 2012; Rabatel et al., 2017). Different methods have been developed to use the end-of-summer snowline observed on air- and spaceborne data, i.e., without direct access to the glacier, to infer glacier mass changes based on a statistical relation between the equilibrium line altitude and the glacier-wide mass balance (e.g., Kulkarni, 1992; Dyurgerov, 1996; Rabatel et al., 2005). These methods were applied to glaciers located in a wide range of different regions, such as in Europe (e.g., Hock et al., 2007; Rabatel et al., 2008, 2016), South America (e.g., Rabatel et al., 2012), New Zealand (e.g., Chinn, 1995,

1999), the Arctic (e.g., Mernild et al., 2013), the Himalayas (e.g., Kulkarni et al., 2004, 2011) and Central Asia (e.g., Dyurg-



erov et al., 1994; Kamniansky and Pertziger, 1996). Some pioneer studies (e.g., Østrem, 1973, 1975; Young, 1981; Dyurgerov et al., 1994) have identified the value of transient snowline observations in connection with subseasonal mass balances. Recent studies (e.g., Hock et al., 2007; Pelto, 2010, 2011; Huss et al., 2013; Hulth et al., 2013) have further developed this concept to improve mass balance monitoring and modelling strategies, including information extracted from continuous snowline obser-

vations. However, most approaches still rely on long-term glaciological information and are thus not applicable to inaccessible glaciers located in remote and unmeasured regions.

In this study, three pillars of a multi-level strategy for glacier observation are combined, covering the period of the past two decades, to improve the understanding of mass change evolution of Abramov, Golubin Glacier and Glacier No. 354, the three benchmark glaciers in the Tien Shan and Pamir-Alay. (1) We integrate in situ glaciological measurements, when available, to

compute annual mass balances using a model-based extrapolation of the measurement points to reach glacier-wide coverage. (2) We calculate geodetic mass changes based on high-resolution digital elevation models (DEMs) on decadal to semi-decadal time scales. (3) We infer daily mass balance series using a model approach supported by transient snowline observations, as a proxy for glacier mass balance. In this way, a temperature index model is calibrated with the snow-covered area fraction (SCAF) of the glacier observed on satellite imagery and time-lapse photographs throughout the ablation season. This approach

represents a new tool for glacier observation at high temporal and spatial resolution. The remote snowline observations provide valuable information, especially for periods for which no direct measurements are available. By combining different independent approaches, we aim to overcome the limitations and shortcomings of each individual method and to deliver a robust mass balance estimate for the three selected glaciers with high temporal resolution for a period for which only limited data has been available so far.

## 20  2   Study Site and Data

### 2.1   Study sites

In this section we present a brief overview of the study sites. A detailed description of the three selected glaciers and their geographic and climatological settings is given in Hoelzle et al. (2017). Table 1 summarizes the available data for each glacier.

#### 2.1.1   Abramov Glacier

Abramov Glacier (N $39°36.78'$, E $71°33.32'$) is located in the Pamir-Alay (North-Western Pamir, Fig. 1). The north to northeast facing glacier has an extent of about $24\,km^2$ (as of 2016) and ranges from 3650 m a.s.l. to nearly 5000 m a.s.l. Barandun et al. (2015) suggested that the glacier had a mean annual balance of $-0.44$ m w.e. $a^{-1}$ between 1968 and 2014 and estimated internal accumulation and basal ablation to contribute by $+0.07$ m w.e. $a^{-1}$ to the total mass change of the glacier. A recent study by Brun et al. (2017) indicates a mass loss of $-0.38 \pm 0.10$ m w.e. $a^{-1}$ for Abramov from ca. 2002 to 2014.

Mean daily air temperature and total daily precipitation sums were measured at a glaciological station located at 3837 m a.s.l. from 1967 to 1998 (Fig. 1). The station was located at a distance of about 0.5 km from the glacier tongue. Air temperature

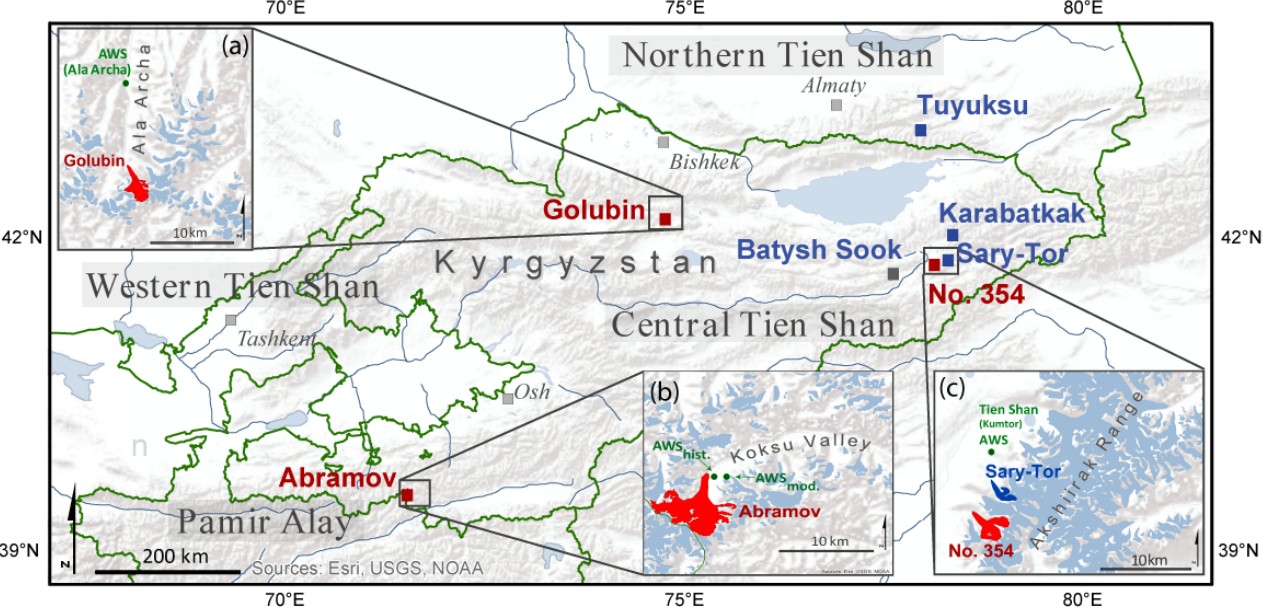

**Figure 1.** Overview map of Central Asia showing the location of glaciers (blue) with available long-term mass balance measurements. Glaciers investigated in this study are marked in red. The insets show the position of the automatic weather station (AWS) for (a) Golubin, (b) Abramov and (c) Glacier No. 354. Note that $AWS_{mod.}$ indicates the new AWS installed in 2011 and $AWS_{hist.}$ the location of the old glaciological station at Abramov.

measured at an Automatic Weather Station ($AWS_{mod}$) installed in 2011 was used from 2011 to 2016 (Fig. 2a). This station is located at an elevation of 4100 m a.s.l. at a distance of about 1.5 km from the glacier terminus. ERA-interim Reanalysis data with spatial resolution of 0.78 degrees (Dee et al., 2011) were used to fill measurement gaps (Barandun et al., 2015).

The glacier mass balance was measured intensively from 1967 to 1998 (Suslov et al., 1980; Glazirin et al., 1993) and
5  the monitoring was re-established in 2011 (Hoelzle et al., 2017). Since then, annual glaciological surveys were continuously carried out in August. For a re-analyzed and reconstructed mass balance series for Abramov from 1968 to 2014 and a detailed description of the measurement network, see Barandun et al. (2015).

### 2.1.2 Golubin Glacier

Golubin Glacier (N42°26.94′, E74°30.10′) is located in the Ala Archa valley in the Kyrgyz Ala-Too in the Northern Tien
10  Shan (Fig. 1). The glacier has an area of ~5 km² (as of 2016) and a north to northwestern aspect. The front is currently located at an elevation of about 3400 m a.s.l. and the glacier spans to an elevation of about 4300 m a.s.l. Long-term measurements indicated an internal accumulation due to refreezing of meltwater of about $+0.08$ m w.e. a⁻¹ (Aizen et al., 1997). For Golubin, mass loss was reported by Bolch (2015) and Brun et al. (2017) for recent decades.



We used meteorological data from the Alplager station located in the Ala Archa valley, situated at an elevation of 2145 m a.s.l. at a distance of about 10 km from the glacier (Fig. 1). There are several other meteorological stations in the valley, however, the Alplager station has the most complete and continuous series at high elevation covering the entire study period.

Intense glacier monitoring started in 1958 and continued until 1994 when the monitoring programme was stopped (Aizen, 1988). In summer 2010, mass balance measurements were re-initiated. Figure 2b summarizes the monitoring network at Golubin Glacier as of 2016, including a mass balance measurement network, an AWS and two terrestrial cameras installed in 2013. A detailed description of the monitoring strategy is provided in Hoelzle et al. (2017).

### 2.1.3 Glacier No. 354

Glacier No. 354 (N 41°47.62′, E 78°9.69′) is situated in the Akshiirak range in the Central Tien Shan (Fig. 1). The glacier covered a surface area of about 6.4 km$^2$ in 2016. The accumulation zone comprises three basins and the glacier tongue is oriented to the northwest. The glacier spans an elevation range of 3750-4680 m a.s.l. Mass loss since the mid-1970s was reported by different studies (Pieczonka and Bolch, 2015; Kronenberg et al., 2016; Brun et al., 2017). Summer snowfall is frequent and fresh snow can cover the entire glacier surface for several days during the melt season, significantly reducing ablation (Kronenberg et al., 2016). Evidence of superimposed ice is found and Kronenberg et al. (2016) estimated internal accumulation to be +0.04 m w.e. a$^{-1}$. An AWS (Tien Shan (Kumtor) AWS) installed at a distance of approximately 10 km to the glacier records continuous meteorological data for the study period (Fig. 1). We used daily precipitation sums and mean daily air temperature for modelling.

Since 2010, in situ mass balance has been obtained annually in late summer (Fig. 2c). Winter measurements exist for May 2014 (Kronenberg et al., 2016). A description of the meteorological input data and the mass balance measurement network, as well as a reconstruction of the mass balance series back to 2003 are provided in Kronenberg et al. (2016).

### 2.2 High-resolution satellite images and DEMs

To compute geodetic mass balances for Abramov Glacier, high-resolution DEMs were used based on Pléiades data acquired in 2015, and on stereo images from 2003 and 2011 from Satellite Pour l'Observation de la Terre (SPOT) 5. For Glacier No. 354, DEMs from 2003 (QuickBird) and from 2012 (GeoEye) were available from Kronenberg et al. (2016). In addition, a SPOT6 stereo-pair acquired in 2015 was used to produce an updated DEM. High-resolution images for Golubin Glacier are sparse and we had to rely on a SPOT7 tri-stereo scene from November 2014 and on Advanced Land Observing Satellite (ALOS) Prism scenes from 2006 (Table 1). For modelling purposes, the most complete and accurate DEM available of each glacier was used to represent the surface topography (Table 1).

### 2.3 Optical satellite and terrestrial camera images

We used freely accessible, orthorectified and georeferenced Landsat TM/ETM+ and OLIs, Terra ASTER-L1B and Sentinel-2A scenes to repeatedly observe the glacier outlines and the snowline throughout the melting season for all three glaciers. In



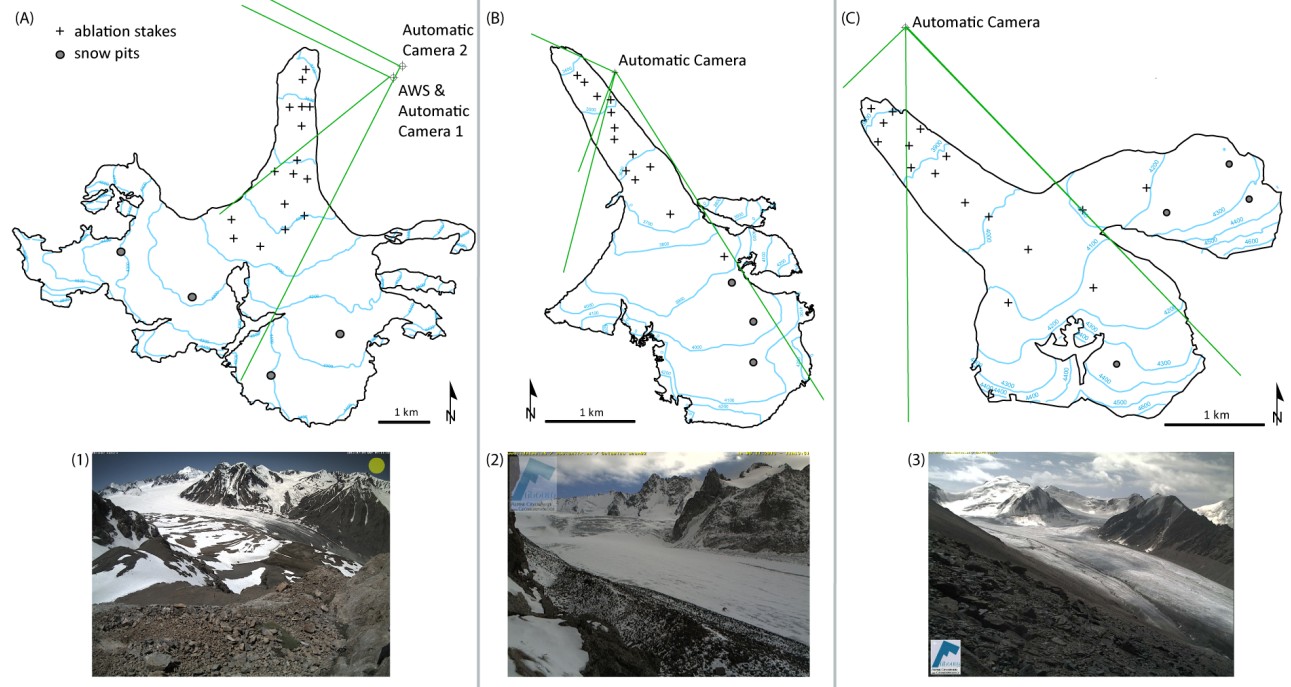

**Figure 2.** Glacier monitoring network at (A) Abramov Glacier, (B) Golubin Glacier and (C) Glacier No. 354. A photograph taken by the terrestrial cameras is shown for (1) Abramov, (2) Golubin and (3) Glacier No. 354.

addition, we used the high-resolution optical satellite images as described above for snowline and glacier outline mapping. Two terrestrial cameras (Mobotix M25) overlooking Abramov Glacier were installed in August 2011. One camera was located next to the $AWS_{mod}$, and the other one at approximately 500 m distance at an elevation of 4200 m a.s.l. (Fig. 2a). Due to multiple camera failures and power supply problems, pictures were lacking from the end of the ablation season in 2012 to the end of

5  the ablation season in 2013, and again partly for the summer months in 2014. In 2015, continuous coverage was obtained from Camera 1 but only a few images could be retrieved from Camera 2 (Fig. 2a). A complete set of data was collected from both cameras for the first time in 2016. A similar setup has been installed for Glacier No. 354 in 2014, delivering continuous coverage since implementation (Fig. 2c). The camera is located at an elevation of 4145 m a.s.l. Images from the two cameras installed at Golubin were not used here due to limited image quality. Figure 3 illustrates the number of camera and satellite

10  images with satisfying quality that were used to obtain maps of snowlines for the three glaciers. Image availability for snowline mapping prior to 1998 was for the most part insufficient.





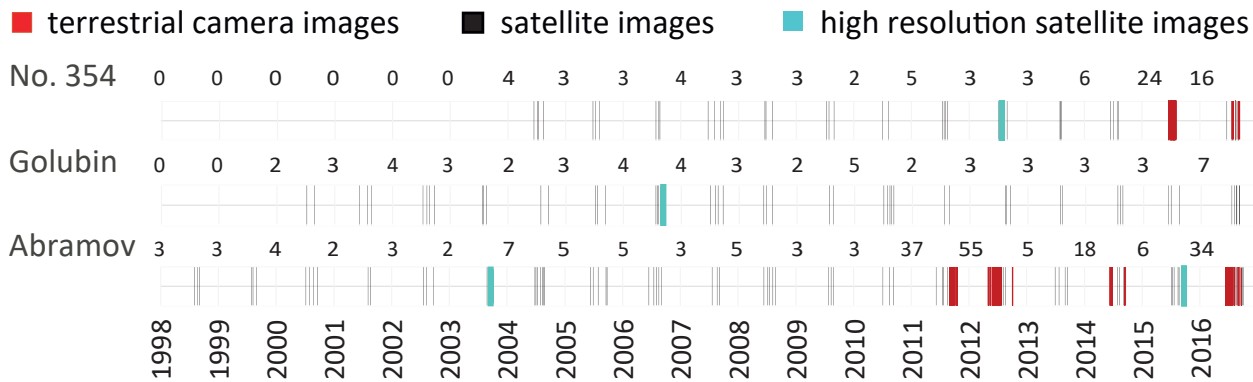

**Figure 3.** Image availability and distribution for snowline mapping. Numbers indicate the total available scenes per year and glacier. Prior to 1998, image coverage is sparse for all three glaciers.

**Table 1.** Available data on glacier monitoring for the three glaciers used in this study. The dates marked with an asterisk indicate the DEMs used for the modelling.

|  | Abramov | Golubin | No. 354 |
|---|---|---|---|
| No. mass balance measurements per year | 22 | 14 | 16 |
| No. annual glaciological surveys (2000 – 2016) | 5 | 6 | 6 |
| Total No. SNL observations | 122 | 56 | 78 |
| High-resolution satellite stereo images | 27/08/2003 | 08/09/2006* | 01/09/2003 |
|  | 29/11/2011 | 01/11/2014 | 27/07/2012* |
|  | 01/09/2015* |  | 01/10/2015 |

# 3 Methods

## 3.1 Glacier outlines

Glacier extents were mapped manually based on satellite images for all three glaciers and for each year of the corresponding study period. Only cloud- and snow-free images were selected. No. 354 and Golubin were mostly debris-free. We excluded a debris-covered part, due to strongly reduced melt rates, at the western margin of Abramov Glacier (Barandun et al., 2015). Annually repeated glacier front measurements using a handheld GPS for all three glaciers were included also for mapping from 2011 onwards. The same extents were used for all three methods. Errors related to glacier outlines digitized manually on remote sensing images depend on atmospheric and topographic corrections, shading, glacier surface characteristics, snow cover and local clouds, but mainly on misinterpretation of debris cover (Paul et al., 2013, 2015). Uncertainties are expected to



be in the range of $\pm 5\%$ of the total glacier area for low-resolution images, and smaller for high-resolution images (Paul et al., 2013).

## 3.2 Meteorological data

For Abramov Glacier, the air temperature data measured at the AWS was adjusted to the elevation of the former glaciological
station by applying a constant lapse rate of $-6\,°C\,km^{-1}$ (Suslov et al., 1980). The ERA-interim Reanalysis dataset was adapted by applying mean monthly additive and multiplicative biases for air temperature and precipitation, respectively. The biases were calculated from long-term in situ measurements (Barandun et al., 2015). From the corrected monthly means, daily series were generated by superimposing day-to-day variability, observed at the meteorological station from 1969 to 1994. For Abramov, we generated air temperature series from 1995 to 2011 and precipitation series from 1995 to 2016. More detailed information
on data preparation and their suitability is given in Barandun et al. (2015). Mean daily air temperature data measured at the Ala Archa AWS and Tien Shan (Kumtor) AWS were extrapolated to the median elevation of the corresponding glacier with monthly temperature lapse rates for the Northern and Central Tien Shan provided in Aizen et al. (1995).

## 3.3 Snowline delineation

A visual pre-selection of suitable camera and satellite images was taken in order to preclude problems associated with image
quality such as fresh snowfall, extensive cloud cover, among others. Oblique ground-based photographs were first corrected automatically for lens distortion, then projected and orthorectified following Corripio (2004). Every pixel on the photograph was associated to the elevation of the DEM. Georeferenced products of satellite scenes were downloaded. On each camera and satellite image, the snow-covered area was digitized manually by means of visual separation of bare ice and snow (Huss et al., 2013; Barandun et al., 2015; Kronenberg et al., 2016). Manual detection allowed the observers knowledge of the snow cover
depletion patterns to be integrated, and was assumed to be less error-prone than an automatic classification (Huss et al., 2013; Rabatel et al., 2013).

Errors occurred due to the pixel size of the images, slope of the terrain, the accuracy of the georeferencing and the quality of the DEM (Rabatel et al., 2012). In view of the fact that the border between ice and snow is not a clearly defined line, operator expertise is desired and beneficial. Contrast becomes rather weak, especially when the snowline rises above the firn
line (Rabatel et al., 2013; Wu et al., 2014). In order to estimate the influence of ambiguous transition areas, we conducted extensive experiments on the interpretation of the surface type (see Section 4).

We assumed the spatial depletion pattern to be approximately constant in time so that camera and satellite images with minor invisible sections of the snowline due to shading, cloud cover, Landsat 7 SLC-off void-stripes or due to the terrestrial camera view angle could be included. To fill in those data gaps, we extrapolated the snowline based on information from repeated
snowline observations of images with good quality over a $\approx$15-year period. The effect of a misinterpretation of the snowline on the calculated mass balance was investigated in detail and is described in Section 4.



### 3.4 Glaciological mass balance

Ablation stakes are distributed over the entire ablation zone in order to provide an optimal representation of melt patterns (Fig. 2). Each year, they are re-drilled at the initial position. An ice density of $900\,\mathrm{kg\,m^{-3}}$ was assigned. Snow pits were dug to the previous end-of-summer horizon to measure snow density and snow accumulation. Annual field surveys ranged from late July to late August and for logistic reasons, can vary from year to year. Winter snow measurements were carried out to retrieve a detailed snow distribution pattern, and to compute the winter balance for No. 354 and Golubin in May 2014 (Kronenberg et al., 2016). Winter surveys from 1993 and 1994 were available for Abramov (Pertziger, 1996). A model-based extrapolation of point measurements to the entire glacier surface after Huss et al. (2009) was used to retrieve glacier-wide mass balances for all years with direct measurements. The model is a combined distributed accumulation (Huss et al., 2008) and temperature index melt model (Hock, 1999) which was automatically optimized to best represent all collected point measurements from each seasonal/annual survey.

### 3.5 Geodetic mass balance

For Abramov Glacier, the 4-m Pléiades DEM from 1 September 2015 was used as reference. It was created using the AMES stereo-pipeline (Shean et al., 2016) and the processing parameters that were used in Marti et al. (2016). The two SPOT5 DEMs (August 2003 and November 2011) were derived from High Resolution Stereoscopic (HRS) images by the French mapping agency (Korona et al., 2009). The steps required to adjust the two SPOT5 DEMs horizontally and vertically to the Pléiades reference DEM are similar to the ones followed in a previous study on the Mont Blanc area (Berthier et al., 2014).

We created DEMs with a spatial resolution of 5 m for Golubin and Glacier No. 354 from the available two/tri-stereo pairs of high-resolution satellite imagery using standard procedures and the software PCI Geomatica (Kronenberg et al., 2016). The two/tri-stereo pairs were connected using common tie points before DEM extraction. For Glacier No. 354 a horizontal shift between the two DEMs was corrected through a DEM co-registration procedure as proposed by Nuth and Kääb (2011). For the data covering Golubin, no horizontal shift was encountered. We thus corrected only for a mean elevation difference of 2.7 m detected over stable ground. For this vertical co-registration, only terrain sections with a slope lower than approximately $30\,^\circ$ were selected and areas with parallax-matching problems or significant snow cover were avoided. Steep mountain walls and shading caused problems. Areas affected by these problems were manually masked out. Unmeasured areas (Abramov: 26% in 2003–2015, and 23% in 2011–2015; Golubin: 30%; Gl. No. 354: 25%) were assumed to have experienced the same elevation change as the measured areas in the same altitude band and the median of the corresponding elevation bin was used for gap-filling. For elevation bins higher than 4300 m a.s.l. at Golubin (9% of total area) and for bins higher than 4500 m a.s.l. at Gl. No. 354 (8% of total area), obvious DEM errors dominated and not enough realistic values for median elevation-change calculation were available. There, the median of the uppermost elevation band with reliable data was used to fill in the gaps.





To derive the geodetic mass balance $\Delta M_{\mathrm{geod}}$, a density $\rho_{\Delta V}$ of $850\,\mathrm{kg\,m}^{-3}$ was used for volume-to-mass conversion (Huss, 2013):

$$\Delta M_{\mathrm{geod}} = \frac{\Delta V \cdot \rho_{\Delta V}}{\overline{A} \cdot \Delta t} \tag{1}$$

where, $\overline{A}$ is the average glacier area and $\Delta t$ is the time in years between the corresponding image pairs. Uncertainties in the
detected elevation changes and the derived geodetic mass balances are described in Section 4.

### 3.6   Mass balance modelling constrained by snowline observations

An accumulation and temperature index melt model closely constrained by transient snowline observations was implemented
in order to infer glacier-wide mass balance. The applied methodology is a further stage in the approach presented by Huss et al.
(2013). The principle of the approach is to employ the information given by the temporal change in the position of the transient
snowline throughout the ablation season to constrain both the amount of winter snow accumulation and melting by iteratively
calibrating a mass balance model. Thus the daily mass balance evolution through each individual year can be inferred. The
approach also allows us to temporally extend mass balance estimates to the end of the hydrological year although snowline
observations do not cover the entire ablation season. The methodological steps are described in more detail in the following.

A mass balance model with a spatial resolution of $20\,\mathrm{m}$ was driven with daily mean air temperature and precipitation sums
measured at nearby meteorological stations or inferred from Reanalysis data (see Section 2.1). We used a classical temperature
index melt model (e.g., Braithwaite, 1995; Hock, 2003). Melt $M$ was calculated for each grid cell $x,y$ and time step $t$ based
on a linear relation with positive daily mean air temperature $T_{\mathrm{air}}(x,y,t)$ as

$$M_{x,y,t} = \begin{cases} DDF_{\mathrm{ice/snow}} \cdot T_{\mathrm{air(x,y,t)}} & T_{\mathrm{air}} > 0° \\ 0 & T_{\mathrm{air}} \leq 0° \end{cases} \tag{2}$$

Daily air temperatures are extrapolated to each grid cell using a constant temperature lapse rate based on literature values
(Table 2). Different degree-day factors $DDF_{\mathrm{ice/snow}}$ were chosen for snow and ice surfaces. The surface type over the glacier
area was given by the snow depth updated with modelled daily snowfall and melt. The ratio between $DDF_{\mathrm{ice}}$ and $DDF_{\mathrm{snow}}$,
$R_{\mathrm{DDF}}$, was held constant over time. As a wide range of different ratios can be found in literature (e.g., Hock, 2003; Zhang
et al., 2006; Gao et al., 2010; Huintjes et al., 2010; Liu and Liu, 2016), we decided to constrain $R_{\mathrm{DDF}}$ by mass balance
measurements in the Tien Shan and Pamir (Table 2). A sensitivity test shows that a variation of $R_{\mathrm{DDF}}$ by $\pm25\%$, a value
exceeding the maximum range found in the literature ($\pm22\%$), only causes small changes in modelled mass balance (Table 3)
indicating that the calibrated $R_{\mathrm{DDF}}$ improves model performance to some extent but is not essential to applying the model. For
more details see Section 4.

Snow accumulation $C$ was calculated for each grid cell $x,y$ and time step $t$ by

$$C_{(x,y,t)} = P_{\mathrm{ws}}(x,y,t) \cdot C_{\mathrm{prec}} \cdot (1 + (z_{(x,y)} - z_{\mathrm{ws}}) \cdot \delta P/\delta z), \tag{3}$$

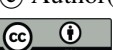



**Table 2.** Constant model parameters. The temperature lapse rate for Abramov Glacier was adopted from Barandun et al. (2015) and for Golubin Glacier and Glacier No. 354 from Aizen et al. (1995). $\delta T/\delta z$, $\delta P/\delta z$, $H_{crit}$ and $R_{DDF}$ are held constant throughout the modelling period. $H_{crit}$ is the elevation where precipitation is set to no longer increase linearly. The initial parameter ranges of $DDF_{snow}$ and $C_{prec}$ as well as the mean value obtained from annual calibration is given with its standard deviation.

| parameter | Abramov | Golubin | No. 354 | unit |
|---|---|---|---|---|
| $\delta T/\delta z$ | $-4.8$ | $-6.3$ | $-6.7$ | $^{\circ}\mathrm{C\,km}^{-1}$ |
| $\delta P/\delta z$ | $6.4$ | $4.5$ | $1.5$ | $\%\,10^{-4}\,\mathrm{m}^{-1}$ |
| $R_{DDF}$ | $1.57$ | $1.36$ | $1.06$ | $-$ |
| $H_{crit}$ | $4400$ | $4000$ | $4500$ | m a.s.l |
| annually variable model parameters | | | | |
| initial range | | | | |
| $DDF_{snow}$ | $3.5-5.5$ | $3.0-5.5$ | $1.5-4.5$ | $\mathrm{mm\,day}^{-1}\,^{\circ}\mathrm{C}^{-1}$ |
| $C_{prec}$ | $1.75-3.0$ | $1.5-3.0$ | $1.0-3.5$ | $-$ |
| best combination | | | | |
| $DDF_{snow}$ | $4.54\pm0.74$ | $5.49\pm0.46$ | $3.04\pm0.66$ | $\mathrm{mm\,day}^{-1}\,^{\circ}\mathrm{C}^{-1}$ |
| $C_{prec}$ | $2.23\pm0.40$ | $1.46\pm0.31$ | $2.35\pm0.29$ | $-$ |

where $P_{ws}$ is the measured precipitation at the meteorological station at elevation $z_{ws}$. $z(x,y)$ is the elevation of each grid cell. The measured precipitation was extrapolated to every grid cell with a constant precipitation gradient $\delta P/\delta z$ calculated from winter snow surveys (Table 2). Solid precipitation occurs at $T_{air} \leq 1.5\,^{\circ}C$ with a linear transition range of $\pm1^{\circ}C$ (e.g., Hock, 1999). $C_{prec}$ is a scaling factor that accounts for gauge under-catch and other systematic measurement errors of pre-
5 cipitation (e.g., Huss et al., 2009). In order to account for smaller measurement errors during summer related to the type of precipitation (solid/liquid, wet/dry snow), $C_{prec}$ was reduced for the summer months to $25\%$ of its value (Sevruk, 1981). Above a critical elevation $H_{crit}$, precipitation is set to no longer increase linearly (Alpert, 1986). Our selected value of $H_{crit}$ approximated the elevation for which a decrease in accumulation was observed on long-term monitored glaciers situated in the Tien Shan and Caucasus (WGMS, 2013). Parameters used are summarized in Table 2.

### 3.6.1 Model calibration

We calibrated $C_{prec}$ and $DDF_{snow}$ keeping $R_{DDF}$ constant. $C_{prec}$ and $DDF_{snow}$ were calibrated annually and for each glacier separately, to correctly represent the winter snow accumulation and the melt rate. To calibrate $C_{prec}$, we relied on the fact that, at the position of the transient snowline, icemelt had not yet started but all winter snow was melted. The modelled cumulative melt, calculated at the position of the observed snowline, is thus interpreted as the total amount of accumulated winter snow
that melted from the onset of the ablation season until the snowline observation date. Using the melt model, we can infer the winter accumulation at the beginning of the ablation season along each observed snowline. This quantity needs to agree with





the directly modelled snow accumulation at the end of the winter season. $DDF_{\mathrm{snow}}$ was calibrated to best represent all SCAF observations of one ablation season (Fig. 4).

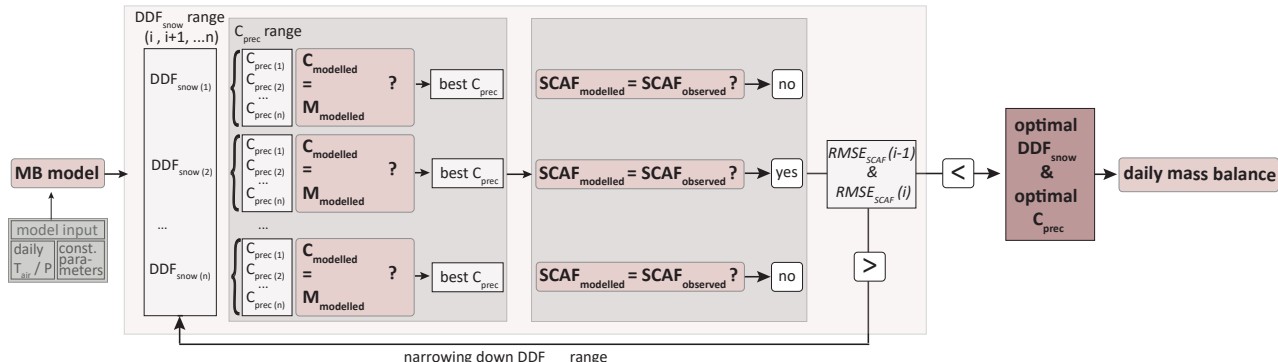

**Figure 4.** Calibration procedure to obtain an ideal combination of $DDF_{\mathrm{snow}}$ and $C_{\mathrm{prec}}$. An initial range for $DDF_{\mathrm{snow}}$ and $C_{\mathrm{prec}}$ was narrowed down through comparison to snowline observations until an optimal solution for both parameters was found. First, for each initial value of $DDF_{\mathrm{snow}}$, the best value of $C_{\mathrm{prec}}$ was determined constraining the modelled cumulative melt ($M_{\mathrm{modelled}}$) at the snowline position to agree with the modelled winter snow accumulation ($C_{\mathrm{modelled}}$) for the same location. Second, the performance of each $DDF_{\mathrm{snow}}$ was evaluated to narrow down the range of $DDF_{\mathrm{snow}}$ by comparing modelled and observed snow-covered area fractions (SCAF). This was repeated until an optimal solution is reached.

As $C_{\mathrm{prec}}$ and $DDF_{\mathrm{snow}}$ depend on each other, overestimation of $DDF_{\mathrm{snow}}$ could cause an overestimation of $C_{\mathrm{prec}}$, and vice versa, without affecting the modelled position of the snowline and the SCAF. To overcome this problem, the values of

$C_{\mathrm{prec}}$ and $DDF_{\mathrm{snow}}$ were constrained to remain within realistic bounds. Through an iterative calibration procedure, we aimed at finding the best possible parameter combination without the need of any additional information.

First, we defined a plausible range for $DDF_{\mathrm{snow}}$ and $C_{\mathrm{prec}}$ for all three glaciers based on literature (Hock, 2003; Liu and Liu, 2016) (Table 2). For each $DDF_{\mathrm{snow}}$, an optimal parameter $C_{\mathrm{prec}}$ was calibrated through iteratively narrowing a plausible range of initial values of $C_{\mathrm{prec}}$ (Fig. 4). In this way, the RMSE between the directly modelled winter snow accumulation

$C_{\mathrm{modelled}}$ and the modelled cumulative melt from the onset of the ablation season to each observation date $M_{\mathrm{modelled}}$ was minimized until no further improvement of the RMSE was observed.

Second, the performance of each $DDF_{\mathrm{snow}}$ with its optimal $C_{\mathrm{prec}}$ pair, was evaluated based on the $\mathrm{RMSE}_{\mathrm{SCAF}}$ between the observed $SCAF_{\mathrm{obs}}$ and the modelled $SCAF_{\mathrm{modelled}}$ for all available snowline observations within one year (Fig. 4). The range of $DDF_{\mathrm{snow}}$ was narrowed around the best solution and the optimization process was restarted until no further significant

improvement of the $\mathrm{RMSE}_{\mathrm{SCAF}}$ was observed. The calibration procedure was repeated for each year individually.

A minimum of two images was needed to enable application of our calibration approach. The influence of the image frequency and distribution was assessed in detail with sensitivity experiments described in Section 4. In a last step, the calibrated model was re-run with the ideal parameter set. This snowline-constrained mass balance model was thus applied to derive con-





tinuous daily mass balance series that agreed with the snow depletion patterns observed by remote sensing imagery. In the following, we refer to the results obtained by the methodology described above as *snowline-derived mass balance*.

### 3.6.2 Adjustments to enable comparison of different methods

Geodetic surveys provide an estimate of the total mass change of a glacier $\Delta M_{\mathrm{geod}}$, whereas snowline-derived mass balance

series refer to the surface balance $B_{\mathrm{a}}$ and do not account for internal and basal components of the mass balance (Cogley et al., 2011). For comparing the results, we adjusted the glaciological and snowline-derived surface mass balances with an estimate of the internal/basal mass balance of $+0.07$ m w.e. a$^{-1}$ for Abramov (Barandun et al., 2015), $+0.08$ m w.e. a$^{-1}$ for Golubin (Aizen et al., 1997) and $+0.04$ m w.e. a$^{-1}$ for No. 354 (Kronenberg et al., 2016). Values are positive due to a significant amount of refreezing of meltwater in cold firn.

To compare the results of the different methods, the time periods covered by the datasets also needed to be homogenized. We thus adjusted the observation period of the snowline-derived mass balance to exactly match the respective periods of geodetic and glaciological mass balance. However, the final results $B_{\mathrm{a(fix)}}$ derived from snowlines are presented for the fixed dates of the hydrological year (1 October to 30 September) and only include internal/basal mass balance.

## 4  Uncertainties and model sensitivity

### 4.1  Glaciological mass balance

Uncertainty $\sigma_{\mathrm{glac}}$ related to the direct glaciological measurements for Abramov was adopted from Barandun et al. (2015), and for Glacier No. 354 from Kronenberg et al. (2016). Uncertainties concerning the glaciological mass balance of Golubin were calculated after Kronenberg et al. (2016). Uncertainties regarding all three glaciers are summarized in Table 3 and range between 0.24 and 0.30 m w.e. a$^{-1}$.

### 4.2  Geodetic mass balance

The total uncertainty of the geodetic mass balance estimate includes a random and systematic error. We followed Brun et al. (2017) for the calculation of the random error on the geodetic mass balance estimate but did not assess the systematic error. Uncertainties in elevation differences were quantified by computing the area-weighted mean of the absolute difference off-glacier in 50 m altitude bins. The resulting values for Abramov, 1.0 m for 2003-2015 and 0.6 m for 2011-2015, and for No. 354,

0.7 m for 2012-2015, are in line with the uncertainty of 1.3 m found over the Mont Blanc area through comparison of similar satellite data to elevation differences measured in situ (Berthier et al., 2014). For Golubin, a value of 1.8 m indicates a slightly lower DEM quality. The uncertainty related to the density assumption for converting volume to mass change was assumed to be $\pm 60$ kg m$^{-3}$ for time intervals larger than five years (Huss, 2013). For shorter periods, we used a more conservative estimate of $\pm 120$ kg m$^{-3}$. The elevation uncertainty for unmeasured glacier zones was roughly estimated to be five times as large than



the uncertainty determined for measured locations. We assumed independence between the different error components and combined them as Root-Sum-Square (RSS) to the total uncertainty for the geodetic mass balance, $\sigma_\mathrm{geod}$.

## 4.3 Snowline-derived mass balance

The uncertainty introduced by the mass balance model constrained by transient snowline observations $\sigma_\mathrm{snl}$ depends on (1) the delineation accuracy of the SCAF, $\sigma_\mathrm{map}$, (2) the image frequency and distribution throughout the ablation season, $\sigma_\mathrm{dis}$, (3) the DEM quality, $\sigma_\mathrm{DEM}$, (4) the meteorological input data, $\sigma_\mathrm{meteo}$, and (5) the uncertainty in constant model parameters, $\sigma_\mathrm{para}$ (Table 3). The individual components were estimated as follows:

(1) The accuracy of the mapped SCAF is dependent on the positioning and the transect of the snowline, the georeferencing of the images, and the extrapolation of the snowline to invisible areas (Huss et al., 2013). The limit between snow- and ice-covered areas is often not a clear line but rather a transition zone, especially for glaciers with superimposed ice. To account for the total uncertainty related to the mapping procedure, we identified an upper- and lowermost position of the surface that could be classified as either snow or ice on each available image. Hence this zone included all ambiguous areas observed, such as cloud-covered regions, shading, superimposed ice or invisibility due to reduced image quality (e.g., Landsat 7 SLC-off void-stripes, invisible areas on photographs). We interpreted the zone to be either entirely snow-covered or entirely snow-free. The standard deviation of the minimal, maximal and optimal SCAF was used as an uncertainty. This uncertainty was calculated for each image individually. To evaluate the corresponding effects on calculated mass balance, the model was re-run with the maximal and the minimal possible SCAF. The standard deviation of the mass balance, $\sigma_\mathrm{map}$, ranged between 0.06 to 0.09 m w.e. $\mathrm{a}^{-1}$ for the three glaciers.

(2) To estimate the effect of varying image availability, we repeated the modelling using different snowline observation frequencies and temporal distributions throughout the summer for calibration. Due to limited image availability, this could only be conducted for the few years when many images were available (Fig. 3). We used the results to create a look-up table that linked the image frequency, the distribution over the ablation season and the last observation date of the season to an uncertainty estimate in the calculated annual mass balance, $\sigma_\mathrm{dis}$. Tests showed that the model reacts more sensitively to the image distribution than to reduced image frequency (Fig. 5). A minimum of two images well distributed throughout the ablation season (i.e., at the beginning/middle and at the end) is sufficient to achieve reliable mass balance estimates. Greater uncertainties were found if images were concentrated on for example, a few days at the beginning of the ablation season (Fig. 5). In this case, image frequency cannot compensate for the missing information on the snow depletion pattern. An image taken towards the end of the ablation season is more important than images from the beginning of the summer. Our assessment of image availability resulted in smaller uncertainties for Abramov ($\sigma_\mathrm{dis} = 0.09$ m w.e. $\mathrm{a}^{-1}$) than for Golubin Glacier ($\sigma_\mathrm{dis} = 0.16$ m w.e. $\mathrm{a}^{-1}$) and for Glacier No. 354 ($\sigma_\mathrm{dis} = 0.18$ m w.e. $\mathrm{a}^{-1}$) (Table 3).

(3) To estimate the uncertainty caused by the DEM used for the modelling, we compared our results to those obtained from model runs that used lower-resolution DEMs. For this experiment, we replaced the high-resolution DEM with the SRTM DEM. This enabled us to both investigate the sensitivity of the results to DEM quality and to assess our assumption of the





unchanged topography for the study period. The effects of a reduced DEM quality for all three glaciers are small ($\sigma_{\mathrm{DEM}} <$ $0.03\,\mathrm{m\,w.e.\,a^{-1}}$).

(4) We investigated the uncertainty related to the meteorological input data, $\sigma_{\mathrm{meteo}}$, by re-running the model with the climatological average daily temperature and precipitation series for each glacier instead of the actual meteorological series.

5  The test assessment revealed an RMSE of $0.13\,\mathrm{m\,w.e.\,a^{-1}}$ for the annual mass balance of Abramov Glacier, of $0.23\,\mathrm{m\,w.e.\,a^{-1}}$ for Golubin Glacier and of $0.14\,\mathrm{m\,w.e.\,a^{-1}}$ for Glacier No. 354. These results demonstrate a relatively low sensitivity of the presented model to meteorological input data and underline the potential of our methodology for regional application based on minimal input data.

(5) To test the uncertainty introduced by the constant (i.e. uncalibrated) model parameters, $\delta T/\delta z$, $\delta P/\delta z$ and the $R_{\mathrm{DDF}}$

10  were varied by $\pm 25\%$ for each glacier and year. A mean standard deviation, $\sigma_{\mathrm{para}}$, of around $0.17\,\mathrm{m\,w.e.\,a^{-1}}$ was found. We additionally tested the behaviour of the model relative to the individual parameters and identified a higher sensitivity to $\delta T/\delta z$ and $R_{\mathrm{DDF}}$, whereas sensitivities to the other parameters were minor (Table 4).

Components 1 to 5 are assumed to be independent of each other and are combined as RSS to represent the total error of the snowline-derived annual mass balance $\sigma_{\mathrm{snl}}$.

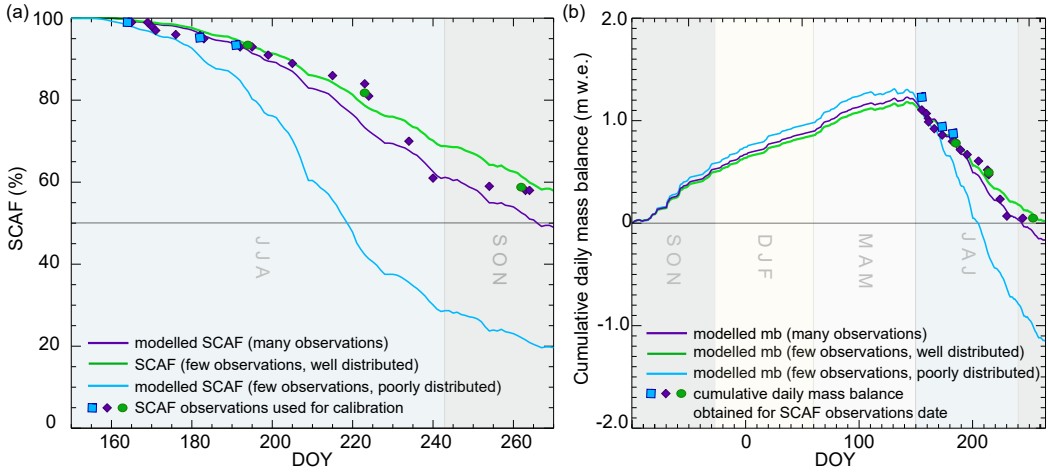

**Figure 5.** Example of the (a) daily SCAF and (b) cumulative daily mass balance (modelled mb) obtained with different sets of image frequency and distribution for Abramov Glacier in 2016. Snowline observation dates used to calibrate the model are indicated with symbols. The modelled daily SCAF and cumulative daily mass balance and their corresponding snowline observations are shown with the same colour. Three images at the beginning of the ablation season (blue), three images well distributed throughout the ablation season (green), and all available images (purple) were used.



**Table 3.** Overall average uncertainties, as well as uncertainties specified for each component related to the geodetic and snowline-derived mass balances for all three investigated glaciers in m w.e. $a^{-1}$.

| Uncertainty in | Abramov | Golubin | No. 354 |
|---|---|---|---|
| **Glaciological mass balance** | | | |
| $\sigma_{\mathrm{glac}}$ | 0.25 | 0.30 | 0.24 |
| **Geodetic mass balance** | | | |
| $\sigma_{\mathrm{geod}}$ | 0.16 (2003–15) | 0.42 | 0.32 |
| | 0.26 (2011–15) | | |
| **Snowline-derived mass balance** | | | |
| $\sigma_{\mathrm{snl}}$ | 0.27 | 0.32 | 0.36 |
| $\sigma_{\mathrm{map}}$ | 0.05 | 0.06 | 0.09 |
| $\sigma_{\mathrm{dis}}$ | 0.03 | 0.16 | 0.18 |
| $\sigma_{\mathrm{DEM}}$ | 0.12 | 0.03 | 0.02 |
| $\sigma_{\mathrm{meteo}}$ | 0.13 | 0.23 | 0.14 |
| $\sigma_{\mathrm{para}}$ | 0.20 | 0.17 | 0.20 |

**Table 4.** Model sensitivity to the different constant input parameters for each glacier in m w.e. $a^{-1}$. See text for details on the experiments.

| sensitivity in | Abramov Glacier | Golubin Glacier | Glacier No. 354 |
|---|---|---|---|
| $\delta T/\delta z$ | 0.06 | 0.14 | 0.05 |
| $\delta P/\delta z$ | 0.07 | 0.08 | 0.01 |
| $R_{\mathrm{DDF}}$ | 0.11 | 0.25 | 0.06 |

## 5 Results

### 5.1 Long-term snowline-derived mass balances

We found that the mass balance model constrained by snowline observations is capable of representing the observed SCAFs on satellite and terrestrial camera images within $\pm 8\%$ for Abramov, $\pm 12\%$ for Golubin and $\pm 7\%$ for Glacier No. 354. Comparing the SCAF observed on camera and on spaceborne images for the same day reveals a RMSE of 2.5%. Tests showed that the influence of the image source (terrestrial/space-borne) on the inferred mass balance is negligible.

Annual glacier-wide, snowline-derived surface mass balance calculated for the hydrological years 1998 to 2016 for the three benchmark glaciers located in the Tien Shan and Pamir-Alay are predominantly negative over the two last decades (Figure 8 and in Table 5). Study periods depend on the data availability for each glacier. Abramov exhibited a mean annual mass balance of $-0.30 \pm 0.19$ m w.e. $a^{-1}$ from 2004 to 2016. For Golubin and Glacier No. 354 a slightly more negative annual average balance of $-0.41 \pm 0.33$ m w.e. $a^{-1}$ and $-0.36 \pm 0.32$ m w.e. $a^{-1}$, respectively, was calculated for the same time period (Table 5). No clear mass-balance trend for the three glaciers was identified over the investigated periods. Two phases of close-to-zero mass





**Table 5.** Annual surface mass balances for the three glaciers for the hydrological year $B_{\mathrm{a(fix)}}$ in m w.e a$^{-1}$ derived from the mass balance model constrained by snowline observations. At the bottom of the table the standard deviation (STD) of the mass balance for each glacier from 2004 to 2016 is given.

| year | Abramov | Golubin | No. 354 |
|---|---|---|---|
| 1998 | $-0.10 \pm 0.21$ | | |
| 1999 | $+0.14 \pm 0.19$ | | |
| 2000 | $-0.69 \pm 0.25$ | $-0.07 \pm 0.47$ | |
| 2001 | $-0.22 \pm 0.21$ | $-0.62 \pm 0.20$ | |
| 2002 | $+0.16 \pm 0.16$ | $-0.13 \pm 0.15$ | |
| 2003 | $-0.31 \pm 0.19$ | $-0.04 \pm 0.51$ | |
| 2004 | $-0.43 \pm 0.22$ | $-0.19 \pm 0.35$ | $-0.33 \pm 0.28$ |
| 2005 | $+0.03 \pm 0.14$ | $-0.03 \pm 0.24$ | $-0.39 \pm 0.24$ |
| 2006 | $-0.59 \pm 0.32$ | $-0.85 \pm 0.35$ | $-0.29 \pm 0.21$ |
| 2007 | $-0.19 \pm 0.18$ | $-0.52 \pm 0.24$ | $-0.40 \pm 0.26$ |
| 2008 | $-0.84 \pm 0.28$ | $-1.42 \pm 0.52$ | $-0.27 \pm 0.47$ |
| 2009 | $+0.07 \pm 0.18$ | $-0.04 \pm 0.43$ | $+0.05 \pm 0.24$ |
| 2010 | $+0.25 \pm 0.17$ | $-0.42 \pm 0.30$ | $-0.22 \pm 0.23$ |
| 2011 | $-0.29 \pm 0.16$ | $+0.26 \pm 0.39$ | $-0.17 \pm 0.39$ |
| 2012 | $-0.65 \pm 0.20$ | $-0.21 \pm 0.23$ | $-0.67 \pm 0.42$ |
| 2013 | $-0.23 \pm 0.16$ | $-0.41 \pm 0.36$ | $-0.41 \pm 0.43$ |
| 2014 | $-0.44 \pm 0.17$ | $-0.62 \pm 0.31$ | $-0.72 \pm 0.55$ |
| 2015 | $-0.25 \pm 0.16$ | $-0.55 \pm 0.24$ | $-0.51 \pm 0.31$ |
| 2016 | $-0.34 \pm 0.17$ | $-0.27 \pm 0.31$ | $-0.29 \pm 0.10$ |
| 2004–2016 | $-0.30 \pm 0.19$ | $-0.41 \pm 0.33$ | $-0.36 \pm 0.32$ |
| STD | 0.29 | 0.40 | 0.19 |

balance could be recognized for 2002-2004 and for 2009-2011 for all glaciers. Glacier No. 354, situated in a more continental climate regime, shows the weakest interannual variability and has a positive balance only in 2009 (Table 5). For Golubin, on the other hand, most negative values were found in 2012 and 2014. Abramov and Golubin had strongly negative balances in 2006 and 2008. For Abramov, the snowline observations indicated that the snowline rose close to the upper edge of the glacier
5 already at the end of August. For Golubin, observations of the last image of the season showed that the SCAF decreased to less than 45% already in mid-August in 2006, similar to 2001 and 2015. The ablation season typically stretched until the end of September, and summer snowfalls during this month were rare (Aizen et al., 1995), likely leading to continued mass loss. Data availability, however, was rather critical for Golubin in 2008, which was also reflected by the stronger uncertainties of the annual mass balance. The last snowline observation dates from as early as the end of July.



## 5.2 Comparison to glaciological and geodetic mass balances

The glaciological and geodetic surveys delivered two extensive and independent datasets for validation of the snowline-derived mass balance series. Joint analysis of the data sets permitted robust conclusions to be drawn about the mass change and its temporal dynamics over the past two decades. The glaciological mass balance measurements showed a good agreement with
the mass balance inferred using snowline observations for the same time periods (Table 6). Annual glaciological mass balances were reproduced with an RMSE of less than $\pm 0.26\,\mathrm{m\,w.e.\,a^{-1}}$ for all three glaciers using the snowline approach (Fig. 6). For Golubin and Glacier No. 354, the glaciological mass balances was slightly more negative than the snowline-derived balances (Table 6). For Abramov, on the other hand, the glaciological mass balance was somewhat less negative than the snowline-derived results. In general, a satisfactory agreement was obtained between the two methods for all three glaciers (Fig. 6).
Squared correlation coefficients between snowline-derived and glaciological mass balances between $R^2=0.63$ (Abramov) and $R^2=0.90$ (Golubin) were found.

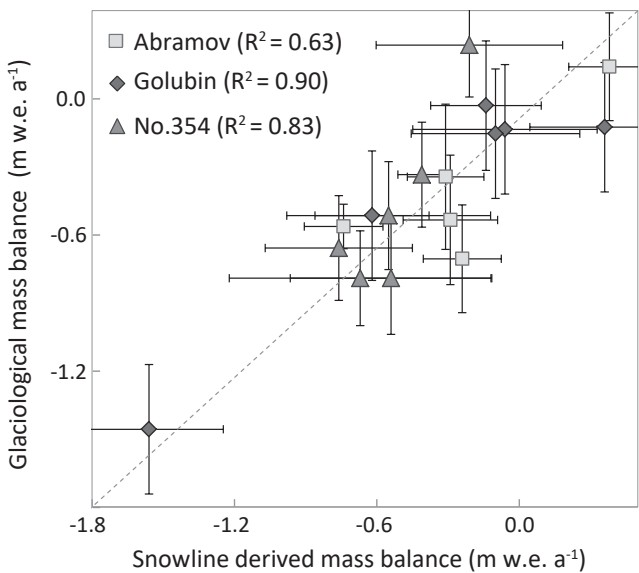

**Figure 6.** Comparison of snowline-derived mass balances and glaciological mass balances for Abramov (squares), Golubin (diamonds), and Glacier No. 354 (triangles). Uncertainties in the annual mass balances are indicated.

Table 7 and Figure 7 summarize the results obtained from the different geodetic surveys. For the comparison with the geodetic mass change, an estimate for internal/basal mass balance was added to the snowline-derived surface mass balance (see Subsection 3.6.2) and referred to as the snowline-derived total mass change. The geodetic method reveals a mass bal-
ance of $-0.22 \pm 0.42\,\mathrm{m\,w.e.\,a^{-1}}$ for Golubin Glacier from 8 September 2006 to 1 November 2014 (Fig. 7). The corresponding total snowline-derived mass balance was more negative with $-0.38 \pm 0.35\,\mathrm{m\,w.e.\,a^{-1}}$ for the same period. However, the





**Table 6.** Annual surface mass balance $B_{a(meas)}$ for the measurement periods (i.e., exact dates of the surveys) based on direct glaciological surveys and on the snowline approach for the three glaciers in m w.e a$^{-1}$.

| | Abramov | | Golubin | | No. 354 | |
|---|---|---|---|---|---|---|
| year | glaciological | snowline-derived | glaciological | snowline-derived | glaciological | snowline-derived |
| 2011 | | | $-0.06 \pm 0.30$ | $-0.05 \pm 0.39$ | $-0.21 \pm 0.27$ | $+0.34 \pm 0.39$ |
| 2012 | $-0.29 \pm 0.30$ | $-0.47 \pm 0.20$ | $-0.14 \pm 0.30$ | $+0.06 \pm 0.23$ | $-0.52 \pm 0.26$ | $-0.74 \pm 0.42$ |
| 2013 | $-0.31 \pm 0.34$ | $-0.27 \pm 0.16$ | $-0.10 \pm 0.30$ | $-0.07 \pm 0.36$ | $-0.54 \pm 0.25$ | $-0.45 \pm 0.43$ |
| 2014 | $-0.74 \pm 0.10$ | $-0.50 \pm 0.17$ | $-1.56 \pm 0.30$ | $-1.44 \pm 0.31$ | $-0.68 \pm 0.22$ | $-0.74 \pm 0.55$ |
| 2015 | $-0.24 \pm 0.25$ | $-0.65 \pm 0.16$ | $-0.62 \pm 0.30$ | $-0.45 \pm 0.24$ | $-0.68 \pm 0.24$ | $-0.60 \pm 0.31$ |
| 2016 | $+0.38 \pm 0.25$ | $+0.24 \pm 0.17$ | $+0.36 \pm 0.30$ | $-0.04 \pm 0.31$ | $-0.41 \pm 0.24$ | $-0.26 \pm 0.10$ |
| 2011–2016 | $-0.24 \pm 0.25$ | $-0.33 \pm 0.17$ | $-0.35 \pm 0.30$ | $-0.33 \pm 0.31$ | $-0.51 \pm 0.24$ | $-0.41 \pm 0.37$ |

**Table 7.** Geodetic mass balance $\Delta M_{geod(meas)}$ and the total annual mass change derived from the snowline approach $\Delta M_{snl(meas)}$ for the three glaciers and for the periods corresponding to the geodetic survey in m w.e a$^{-1}$.

| | $\Delta M_{geod(meas)}$ | $\Delta M_{snl(meas)}$ |
|---|---|---|
| Abramov | | |
| 2003–2015 | $-0.39 \pm 0.16$ | $-0.25 \pm 0.20$ |
| 2011–2015 | $-0.36 \pm 0.26$ | $-0.43 \pm 0.17$ |
| Golubin | | |
| 2006–2014 | $-0.22 \pm 0.42$ | $-0.38 \pm 0.35$ |
| No. 354 | | |
| 2003–2012 | $-0.42 \pm 0.07$ | $-0.25 \pm 0.31$ |
| 2012–2015 | $-0.58 \pm 0.31$ | $-0.53 \pm 0.43$ |

results of the two methods agree within their error bars (Table 7 and Fig. 9). Comparison of digital elevation models indicated that Glacier No. 354 had a mass balance of $-0.58 \pm 0.31$ m w.e. a$^{-1}$ from 27 July 2012 to 1 October 2015 (Fig. 7) and $-0.42 \pm 0.07$ m w.e. a$^{-1}$ from 1 September 2003 to 27 July 2012 (Kronenberg et al., 2016). The total annual mass change for the same time intervals derived from the snowline approach was $-0.53 \pm 0.43$ m w.e. a$^{-1}$ and $-0.25 \pm 0.31$ m w.e. a$^{-1}$,

5  respectively. For the first period, the results are in good agreement, whereas for the second period the mass balance model constrained by snowline observations indicates a significantly less negative mass balance (Fig. 9). For Abramov, a geodetic mass balance of $-0.39 \pm 0.16$ m w.e. a$^{-1}$ from 27 August 2003 to 1 September 2015 and of $-0.36 \pm 0.26$ m w.e. a$^{-1}$ from 29 November 2011 to 1 September 2015 was calculated. For the same periods the snowline model reveals a total mass change of $-0.25 \pm 0.20$ m w.e. a$^{-1}$ and of $-0.43 \pm 0.17$ m w.e. a$^{-1}$, respectively. For the first period, the snowline-derived mass balance indicates a less negative mass balance. The second period is in good agreement (Table 7 and Fig. 9). For all three glaciers and

10  periods studied the differences are within the error margins.

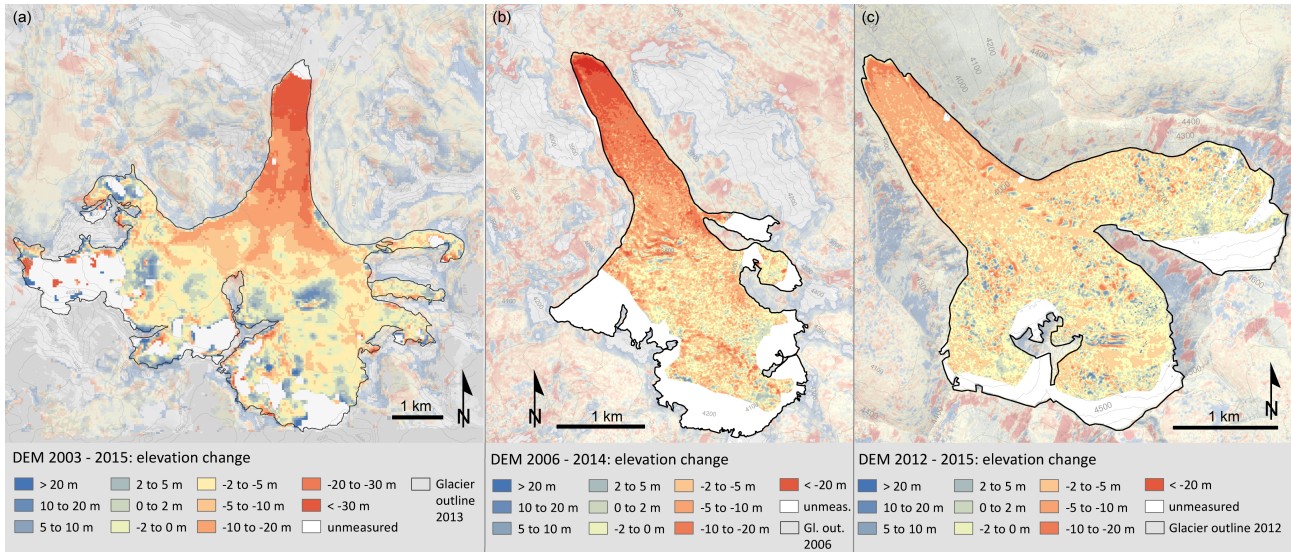

**Figure 7.** Geodetic mass balance for (a) Abramov Glacier from 2003 to 2015, (b) Golubin Glacier from 2006 to 2014 and for (c) Glacier No. 354 from 2012 to 2015.

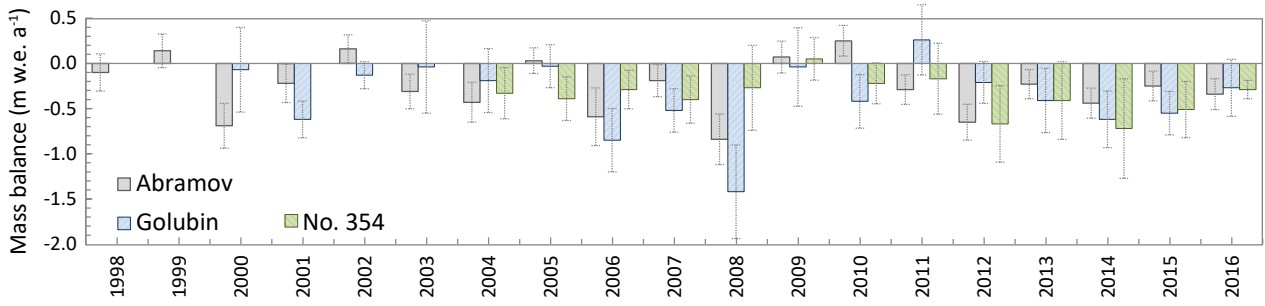

**Figure 8.** Snowline-derived annual surface mass balance for all three glaciers for the hydrological year.

## 6 Discussion

### 6.1 More accurate modelling through integrating snowline observations

In order to demonstrate the advantage of integrating snowline observations on repeated remote sensing data throughout the melting season for increasing the confidence in mass balance modelling, we ran the same accumulation and temperature-
5  index model without the use of snowline or any other direct observations for calibration for all three glaciers from 2004 to 2016 (See Section 3.6). The same constant parameters were used (Table 2). We chose $C_{\text{prec}}$ to account for a 20%-measurement error of the recorded precipitation (Sevruk, 1981), and a combination for $DDF_{\text{ice}}$ (7.0 mm day$^{-1}$ °C$^{-1}$) and





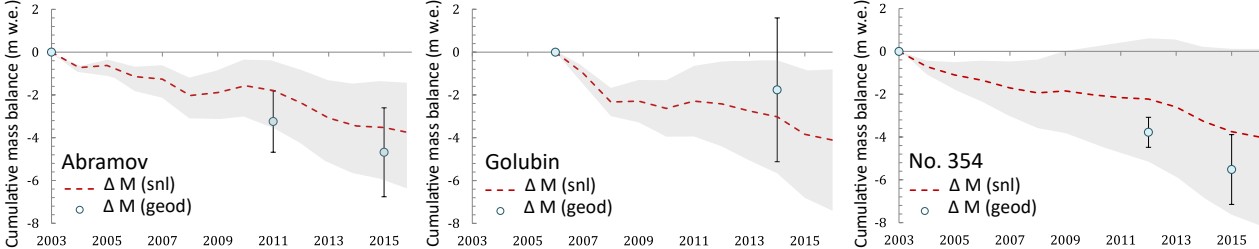

**Figure 9.** Cumulative snowline-derived mass change (red) in comparison to the geodetic mass change (circles) for (a) Abramov, (b) Golubin and (c) Glacier No. 354. The shading indicates the uncertainty range of the snowline-derived mass change.

$DDF_{snow}$ (5.5 mm day$^{-1}$ °C$^{-1}$) as recommended by Hock (2003) for the former Soviet territory, for all three glaciers. The parameters were held constant over time. Figure 10 shows the difference between the cumulative mass balance derived from our model constrained by snowline observations, and the results obtained with an unconstrained mass balance model. The results clearly indicate the potential of the snowline approach to infer mass balance series of unmeasured glaciers without any additional information. The unconstrained mass balance model overestimates mass loss by roughly ten times for Abramov and about five times for Golubin, whereas mass loss for Glacier No. 354 is strongly underestimated (Fig. 10).

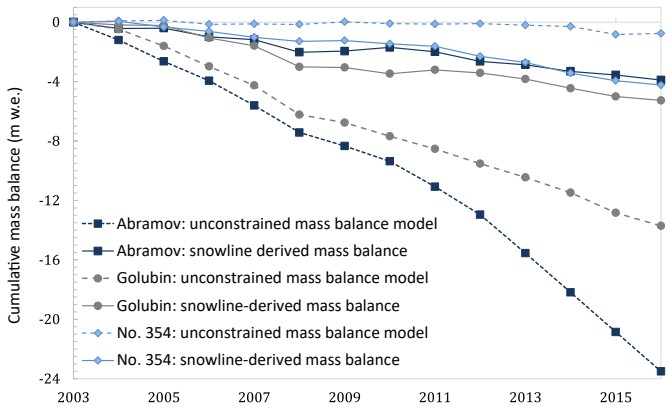

**Figure 10.** Comparison of the cumulative mass balance derived from unconstrained mass balance modelling to snowline-derived mass balance series from 2004 to 2016.

## 6.2    Intercomparison of methods to determine glacier mass balance

A satisfying agreement was found between all three independent methods used to compute glacier-wide mass balance for the three benchmark glaciers in the Central Tien Shan and Pamir-Alay for the past two decades. In the following we discuss shortcomings and advantages of each method and point out the limitations of the individual approaches.

The snowline model represents the direct measurements well and performs satisfactory for all three glaciers (Table 7, 5 and Fig. 11). For Glacier No. 354, a larger misfit between the glaciological balance and the snowline model was found, especially for 2011. A possible reason might be the limited stake network at the initiation of the monitoring programme, large errors of the stake readings and an unknown measurement date for some ablation stakes, affecting the calculation of the glaciological

mass balance for the considered year. Discrepancies between the two approaches are also found at Golubin for 2016 and at Abramov for 2015. For Abramov, summer snowfalls in mid-August 2015 stopped the ablation season early. This pattern is mirrored in the daily mass balance derived from the snowline approach. For Golubin, the last snowline observation is from the end of August and matches the field observations. No clear indication of a poor performance of the snowline approach could thus be identified for both of the glaciers for the considered years.

An important problem is the varying measurement periods of the glaciological mass balances for the selected glaciers. Due to changing period lengths, the data do not always represent a complete mass-balance year, and might thus not be representative, making comparison of the results with other methods, glaciers and regions difficult. Interpretation of the results, their contextualization and application in other study fields, such as in hydrology or climatology, are also hampered through the varying and irregular investigation periods. Based on our methodology, we are now able to derive homogenous glacier mass

balances for comparable periods of the hydrological year.

An important factor limiting the applicability of mass balance modelling constrained by snowline observations is the dependence on good satellite imagery to map the snowline throughout the ablation season. However, the sensitivity analysis (Section 4) shows that a minimum of only two images that are well distributed throughout the ablation season are sufficient to retrieve reliable results. Image availability is most important close to the end of the ablation season. Taking into account the increasing

number of satellite sensors that provide a range of possibilities to observe snowlines in the future will partly resolve this limitation. By comparing the snowline-derived daily mass balance for the years in which seasonal in situ measurements are available, we were able to investigate the effect on the results of the two parameters used for model calibration. Figure 11b shows that the mass balance of Golubin is slightly underestimated at the beginning of the ablation season, and hence the modelled melt is also too low. This shortcoming of the calibration procedure cannot be overcome without including additional data.

The geodetic mass balance and the snowline-derived results agree well, in particular for the recent years (Fig. 9). Overall, a slightly greater mass loss is calculated for Abramov and Glacier No. 354 using DEM differencing. Especially during the earlier part of our study period, the mass balance inferred with the snowline approach seems to be too positive. Limitations related to the geodetic approach are mainly connected to the limited stereo acquisitions in the first years of the 21st century. In recent years, image availability strongly increased, but it is still not common to find a suitable scene from the end of the

hydrological year for a selected glacier or region with sufficient quality for a sound geodetic evaluation. Fresh snowfall or low image contrast (in particular in the accumulation areas) interfere with the DEM quality but cannot be avoided and have thus to be corrected for, increasing the uncertainty of the result. We present geodetic mass balances for periods shorter than five years, a critical time interval for an accurate volume-to-mass conversion. Huss (2013) shows a high variability of the volume-to-mass conversion factor for short observation periods ($\leq 3$ years), especially for glaciers with close-to-zero mass balances in

combination with strongly varying mass balance gradients. For the observation periods considered in this study, annual mass



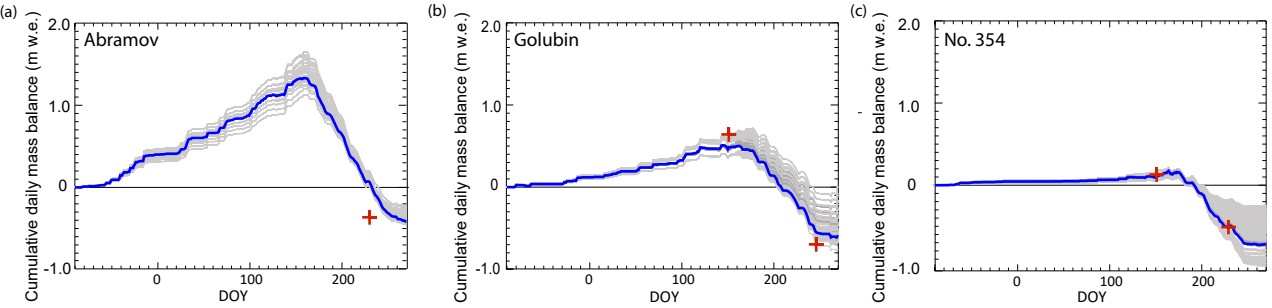

**Figure 11.** Cumulative daily glacier-wide mass balance inferred from the snowline approach for (a) Abramov, (b) Golubin and (c) No. 354 for the year 2014 (blue line). The grey lines indicate the spread obtained by using different constant parameters to run the model (Section 4) and the red cross indicates the measured glaciological balance both for the winter and the annual period.

balances were predominantly negative but moderate variations of the mass balance gradients have been observed (Barandun et al., 2015; Kronenberg et al., 2016). We identified a rather large elevation change for the short observation periods, and are thus confident that the chosen conversion factor lies within the uncertainty range assigned here (see Section 4).

The glaciological and the snowline-derived mass balances refer to surface balance components only. The geodetic mass bal-
ance, on the other hand, takes into account the total mass change, thus including internal and basal ablation and accumulation. This is a limiting factor for direct comparison. Evidence of refreezing meltwater in cold firn is reported for all three glaciers (Suslov et al., 1980; Aizen et al., 1997; Dyurgerov and Mikhalenko, 1995) and can have a significant effect on the total mass change. The values used in this study to account for internal and basal mass balance are first-order approximations which improve the comparability between the different methods. However, the uncertainties in these estimates are considerable.

## 6.3    Comparison to other studies

We performed a comprehensive comparison of long-term averages of mass balance derived from the snowline approach to independent studies based on geodetic surveys using different sensors and modelling, both for the investigated glaciers and for regional mass balances (Fig. 12). Note that the study periods vary between the different studies and results might thus not be directly comparable.

For Abramov Glacier, we find mass balances in between the results derived by Gardelle et al. (2013) and Brun et al. (2017) based on the comparison of DEMs, overlapping within the respective uncertainty ranges. Mass changes reported by Gardelle et al. (2013) are most likely too positive as SRTM C-Band penetration depth into snow (Kääb et al., 2015; Berthier et al., 2016) might have been underestimated for the cold and dry snow of accumulation areas (Dehecq et al., 2016). The average mass balance for Abramov of $-0.38 \pm 0.10 \, \mathrm{m \, w.e. \, a^{-1}}$ (2002-2014) derived by Brun et al. (2017) using multi-temporal ASTER
DEMs indicates a stronger mass loss than our study. We note, however, that the start and end dates of their geodetic mass





balance represents a mean over a mosaic of different dates, thus hampering the direct comparison to our results; in addition the differences are still within their error bounds.

For Golubin Glacier, the inferred mass balance is in close agreement with the geodetic mass change reported by Bolch (2015) (Fig. 12b). Brun et al. (2017) computed a mass balance of $-0.04 \pm 0.19$ m w.e. a$^{-1}$ for $\approx$2002-2013, which is con-

sistently less negative than our estimate but still lies within the respective error bounds. For Glacier No. 354, an excellent agreement between all available mass balance assessments was found (Fig. 12c). Brun et al. (2017) reported a mass balance of $-0.46 \pm 0.19$ m w.e. a$^{-1}$ for $\approx$2002-2014.

Furthermore, we also compared our results for the investigated glaciers to region-wide assessments in order to investigate their regional representativeness (Fig. 12d-f). Brun et al. (2017) divided the Pamir-Alay and the Pamir into two different

regions, whereas Gardner et al. (2013), Gardelle et al. (2013), Kääb et al. (2015) and Farinotti et al. (2015) did not make this distinction. For the Pamir, widely varying mass balance estimates were presented by the different studies which might be related to the important methodological differences and inconsistent time periods considered. Our results for Abramov are close to the average of the regional studies (Fig. 12d). The interannual variability and in particular a very negative mass balance for 2008 and a positive balance in 2010 for Abramov found in the present study agrees well with modelled mass balance series

reported by Pohl et al. (2017) for the Pamir from 2002 to 2013. The snowline-derived mass balance for Golubin agrees with the region-wide estimates by Brun et al. (2017) but indicates smaller mass losses than other large-scale studies (Fig. 12e). Close agreement between the different regional studies is found for the Central/Inner Tien Shan where Glacier No. 354 is located (Fig. 12f).

## 7   Conclusions

In this study we used three independent methods to reconstruct robust mass balance series at high temporal resolution for Abramov, Golubin and No. 354 located in the Pamir-Alay and Tien Shan mountains for the past two decades – a period for which only little is known about glacier behaviour. We proposed a methodology to derive glacier surface mass balances series for unmeasured glaciers based on mass balance modelling constrained by repeated snowline observations. We recommended including snowline observations in the glacier monitoring strategy to reduce uncertainty and to increase robustness of the

data. We used extensive geodetic and glaciological surveys to validate the results, and found satisfying agreement between the independent methods. Our snowline approach reproduced observed annual to decadal mass balances satisfactorily for all three glaciers, and enabled the calculation of daily mass balances for arbitrary periods, and is, hence, capable of covering the entire hydrological year based on minimal data input. Some of the shortcomings of the glaciological and geodetic surveys could thus be overcome.

The results of all three methods confirm a continuous mass loss of the three benchmark glaciers Abramov, Golubin and No. 354 for the past two decades but no clear mass balance trend could be identified for the time period considered. Our results suggest a slightly less negative surface mass balance for Abramov of $-0.30 \pm 0.19$ m w.e. a$^{-1}$ located in the Pamir-Alay than for the Tien Shan glaciers Golubin of $-0.41 \pm 0.33$ m w.e. a$^{-1}$ and Glacier No. 354 of $-0.36 \pm 0.32$ m w.e. a$^{-1}$ from 2004 to



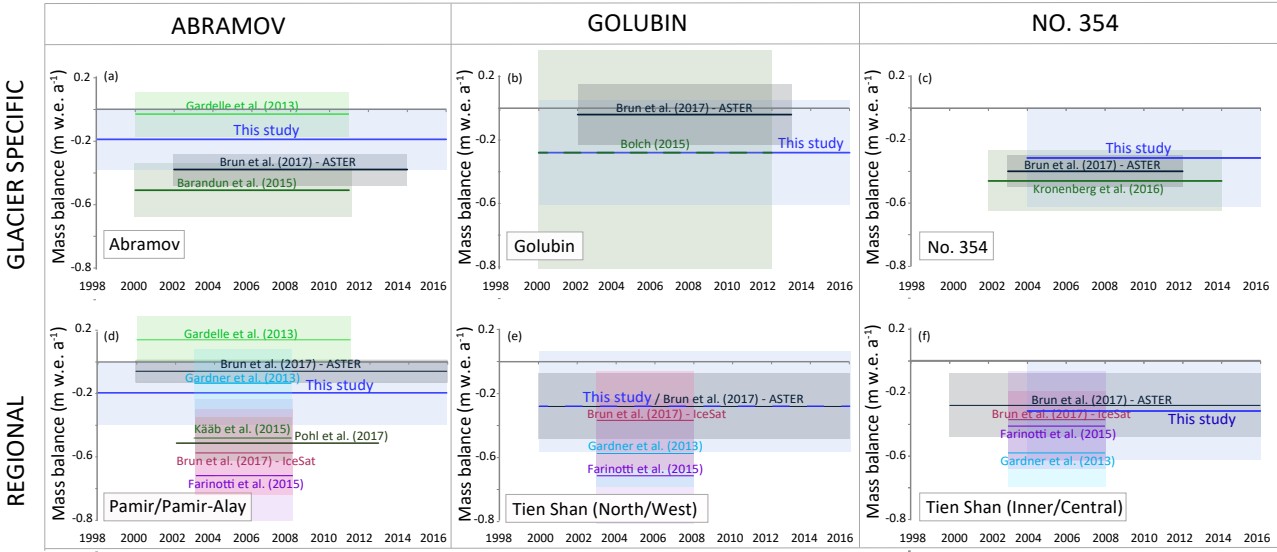

**Figure 12.** Long-term average mass balances from various studies in comparison with the snowline-derived mass balance for (a) Abramov, (b) Golubin, (c) Gl. No. 354. Results from the snowline approach for the three glaciers are compared to region-wide mass balance estimates for (d) the Pamir, for (e) Tien Shan (North/West) and for (f) the Tien Shan (Inner/Central). Lines indicate the mean annual mass balance over the respective periods for each study, and shaded squares represent the corresponding uncertainty. For the study by Brun et al. (2017) results based on ASTER DEMs and IceSat are presented separately.

2016. Periods of almost balanced mass budgets were observed from 2002 to 2004 and from 2009 to 2011. The mass balance of 2006 to 2008 was very negative for Abramov and Golubin. Glacier No. 354 shows a weaker inter-annual variability than the other two glaciers, explained by its more continental climate regime. Model sensitivity experiments revealed a relatively small sensitivity to the input parameters and the meteorological data used, indicating a considerable advantage in comparison

5  to conventional mass balance modelling that does not include direct glacier-specific observations. Our results show that with a minimum of two snowline observations, ideally at the beginning and the end of the ablation season, reliable estimates of the annual mass balance can be inferred.

At present, mass balance observations in the Pamir and Tien Shan mountains are sparse but crucially needed to improve understanding of glacier behaviour in the region and its effect on future water availability. Direct measurements are, how-

10  ever, costly and laborious and require an immense logistic effort. For remote and inaccessible regions and countries, lacking in financial resources and infrastructure to support such monitoring programmes, our proposed approach delivers a tool for investigating and reconstructing the mass balance of inaccessible and remote glaciers with minimal effort. The integration of snowline observations into conventional modelling is shown to be highly beneficial, for filling the gaps in long-term mass balance series for periods for which direct glaciological measurements were discontinued or are missing completely.



*Acknowledgements.* This study is supported by the Swiss National Science Foundation (SNSF), grant 200021_155903. Additional support by the German Federal Foreign Office in the frame of the CAWa project (http://www.cawa-project.net) and the support of the Federal Office of Meteorology and Climatology MeteoSwiss through the project Capacity Building and Twinning for Climate Observing Systems (CATCOS) Phase 1 & 2, Contract nos. 7$F$-08114.1, 7$F$-08114.02.01, between the Swiss Agency for Development and Cooperation (SDC)

5  and MeteoSwiss as well as the project CICADA (Cryospheric Climate Services for improved Adaptation), and contract no. 81049674 between Swiss Agency for Development and Cooperation and the University of Fribourg is equally acknowledged. E. Berthier acknowledges support from the French Space Agency (CNES) and the Programme National de Télédétection Spatiale grant PNTS−2016−01. A. Kääb acknowledges funding by ESA (Glaciers_cci project 4000109873/14/I-NB), and the European Union Seventh Framework Program (FP7) under European Research Council (ERC) contract 320816. We thank F. Brun for providing the elevation change data of Abramov, Golubin

10  and Gl. No. 354 from 2002–2013. J. Corripio is acknowledged for the software to georeference oblique photographs. We extend our thanks as well to T. Saks, A. Ghirlanda, A. Gafurov, M. Kronenberg, D. Sciboz and all others who contributed with fieldwork. We are also grateful for the collaboration of Central Asian Institute for Applied Geosciences, especially to B. Moldobekov for his continuous support. The Kumtor Gold Company provided the meteorological data. We thank S. Braun-Clarke for the proofreading and linguistic revision.



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
