# Peer review of "Multi-decadal mass balance series of three Kyrgyz glaciers inferred from modelling constrained with repeated snowline observations"

_The Cryosphere, 2017_

## Referee Comment (RC1) · Anonymous Referee #1 · 22 Dec 2017

Overall comment:

This paper by Barandun and colleagues presents an interesting approach to model the glacier surface mass balance using a degree-day approach for ablation, precipitations for accumulation and remotely-sensed snowline observations to constrain the model.

This approach is applied on three glaciers located in Kyrgyzstan where glacier surface mass balance data from the glaciological and geodetic methods are available to validate the modeling approach.

The approach is also used to reconstruct glacier surface mass balance time series over a longer time period than available in situ glaciological data.

[Figure]

The paper reads well. However, I have some comments, and I consider that once integrated by the authors, the paper will be ready to be published in The Cryosphere.

=================================

Main comments:

(1) Title of the paper

The title of the paper is misleading and needs to be changed. Indeed, in your approach the surface mass balance is modeled with meteorological data and this modeling is constrained by snowline observations during the ablation season. Thus, the title has to mention that the surface mass balance time series are quantified with a model. As it stands, one can have the impression that the surface mass balance is only inferred from snowline observations.

I suggest:

"Glacier surface mass balance modeling constrained by remote sensing derived snowlines. Application on three kyrgyz glaciers to quantify multi-decadal series".

or

"The use of remote sensing derived snowlines to constrain glacier surface mass balance modeling. Application on ..."

(2) Name of the proposed approach.

In the same way, the name you give to the proposed approach (snowline-derived mass balance) is not adequate, and so is the abbreviate in the tables and figures (Msnl).

For the name of the approach, I propose: Modeled SMB constrained by snowline.

And for the abbreviate: Mmod-tsl. TSL stands for Transient Snow Line and must be preferred to SNL.

(3) Terminology. "Mass balance", "glacier balance", etc. . .

Be careful with the terminology and follow Cogley et al. Glossary of mass balance terminology. For example, use "surface mass balance" instead of "mass balance" or "glacier balance", except when you refer to the geodetic mass balance.

(4) Glaciological surface mass balance.

The used method is not the classical glaciological one. Indeed, a model-based extrapolation is used and optimized to best fit the point measurements. What about the input data for accumulation and ablation (e.g. precipitation and temperature data)? Where do they come from? Are they the same as used in the model constrained by the snowline?

If the data are the same, are the methods "constrained by the snowline observations" and "constrained by the point measurements" strictly independent as they rely on the same meteorological data?

(5) High resolution images recorded in November => impact of the snow cover on the derived DEM.

Two of the stereo-pairs used to make DEMs and quantify the geodetic mass balance were recorded in November for Abramov and Golubin glaciers, respectively.

I had a look at the SPOT catalogue and for Golubin I could find the SPOT7 images from 01/11/2014. Regarding Abramov, the images are not in the catalogue but I assume it is because these have been acquired within the SPIRIT acquisition campaign.

Anyway, at least for 2014 and I assume it is the same for 2011, all the more because the SPOT5 images date from late November, the glaciers and surrounding terrains are completely snow covered.

This implies several challenges for DEM generation:

- low contrast because of snow brightness

- unknown snow thickness

- impossibility to delineate the glacier outline.

How these issues have been tackled and what is their impact on the uncertainties?

In addition, Table 7 shows that the geodetic mass balances for Abramov (2011-2015) and for Golubin (2006-2014) are less negative that the average annual surface mass balances quantified by your model constrained by the snowline. Can the snow-cover on the images implying a higher surface elevation (of an unknown value!) be a cause of this difference (or at least part of it)?

(6) Surface mass balance interannual variability.

It is pity that your discussion about the interannual variability of the SMB is short and only dedicated on the annual values.

You should have a look on the two terms of the annual surface mass balance, and discuss about their interannual variability. You could see if the interannual variabilities of accumulation and ablation are homogeneous, comparable between the three sites and if the difference you mention regarding the interannual variability of Glacier No. 354 annual SMB is more likely related to a different ablation or accumulation interannual variability.

(7) Application of the proposed approach.

You mention that the approach you propose can be useful to quantify surface mass balance time series for a number of remote glaciers. Although I mostly agree with this statement, I wonder if the approach can indeed be applicable for summer accumulation glaciers (like in the tropics, or in monsoon regime regions) or for high latitudes glaciers where superimposed ice can be more important than in Kyrgyzstan.

You can probably add some sentences on this point in the conclusion.

=======================================

Specific comments (PX, LX = Page X, Line X)

Abstract

- P1, L5-6: the sentence "A combination of 3 independent [...] Golubin and No. 354" needs to be reformulated. Indeed, the methods are not combined to reconstruct the surface mass balance. The methods are compared/cross-validated but not combined. For me, you would have a combination if, for example, your modeled surface mass balance time series had been adjusted with the geodetic method.

- P1, L8-9: "satellite optical imagery" instead of "satellite imagery".

- P1, L12, 13 and in the entire paper: should write "yr-1", instead of "a-1". "yr-1" is most common except for IGS journals.

- P1, L15: prefer "unmonitored" to "inaccessible". I do not know about an inaccessible glacier on Earth.

1. Introduction

- P2, L2: should the IPCC reference be quoted Stoecker et al., 2013?

- P2, L18: remove a comma after "e.g.,"

2. Study site and data

- P4, L12-13: this statement is useless if you do not give any quantification.

- P5, L11-12: idem, give a value.

- P5, L6 and 15: should mention the elevation of the AWS.

- Figure 3: for Glacier No. 354, the Quickbird and Pléiades images recorded in 2003 and 2015 respectively have not been used for snowline mapping? If yes, these images should appear in the Figure. If not, why?

- Table 1: should write "snowline" instead of SNL. You should also indicate the sensor in brackets for the high resolution images. If I am correct these are SPOT5 and Pléiades for Abramov, ALOS and SPOT7 for Golubin and Quickbird, GeoEye and Pléiades for

No. 354.

3.4 Glaciological mass balance

See my main comment related to this section (#4), and the one regarding the terminology for the title of this sub-section.

3.5 Geodetic mass balance

See my main comment related to this section (#5).

3.6 Mass balance modeling. . .

- P10, L23: the paper by Huintjes et al. has not been finally published. I wonder if papers that stayed in discussion and/or were rejected can be quoted. You can probably remove this reference from the text and the ref list, all the more that it is quoted within a list of 5 references starting by e.g.

- In Table 2 and in the text (e.g. P11, L7): I recommend using 'Z' instead of 'H' for the elevation criteria. H usually stands for thickness.

4.3 Snowline-derived mass balance

- P15, L3-8: you should provide an illustration for the test that uses average daily temperature and precipitation series. In addition, you have to explain the low sensitivity of your model to the input meteorological data. I assume this is because the parameters are adjusted for each year and for each glacier.

- Figure 5b: change JAJ by JJA.

5.1 Long-term snowline. . .

- P16, L3-6: you should provide an illustration for the comparison between SCAFs given by the model and the images for all the used images, glacier by glacier.

- P16, L7-8: the period 1998 to 2016 stands for Abramov Glacier only. You have to mention the specific periods for each glacier. Golubin starts in 2000 and No. 354

in 2004. In addition "over the two last decades" can be removed from the sentence because the time periods for each glacier will be mentioned.

- P16, L8: you refer to Figure 8, but because figures 6 and 7 have not been quoted yet, this figure should be Figure 6. Anyway, because I suggest adding two more figures, it will probably remain Figure 8, but must appear before the current figures 6 and 7.

- Table 5: you must indicate in the table itself (not only the caption) on which period the STD is quantified.

- P17, L1: should the first close-to-zero SMB period be extended to 2005?

- P17, L1: "Glacier No. 354, situated in a more continental climate regime, [...]". I am a bit skeptical with this statement! All three glaciers are in a continental regime. This glacier being located in the inner range, it might receive less precipitation than the two others. Is it what you mean? See also my main comment related to interannual variability of the SMB (#6).

- P17, L4-5: this sentence refers to the years 2006 and 2008 mentioned in the previous sentence? If yes, the two sentences might be separated by a semi-colon not a dot.

- P17, L8-9: same thing here, the two sentences could be separated by a semi-colon not a dot.

5.2 Comparison to glaciological...

- Table 6: you should mention in the caption that the values differ from Table 5 because the glacier-wide annual surface mass balances are not computed over the same number of days. However, the difference is really high for some years, for example 2014 for Golubin Glacier (more than 0.8 m w.e. different!). You could indicate the number of days differing from the quantification given in Table 5.

- Table 7. A dot is missing after 'e' in "m w.e a-1" In addition, regarding Abramov Glacier, why the first period is 2003-2015 and not 2003-2011?

6.1 More accurate. . .

- P20, L3: replace "integrating" by "using"

- P21, L6: discuss why the difference is opposite for Glacier No. 354

6.2 Intercomparison. . .

- P22, L27: replace "too positive" by "not negative enough"

- P22, L33: change "shows" by "showed"

6.3 Comparison to other studies

- P23, L18-20: you indicate that the average glacier-wide annual surface mass balance quantified by Brun et al. (2017) shows a stronger mass loss than your study. This is not really exact. The difference is important with your modeling approach, but the estimate by Brun et al. and your geodetic estimate are really close.

---

## Referee Comment (RC2) · M. Pelto (Referee) · 23 Dec 2017

Barundun et al (2017) provide a valuable reconstruction of the mass balance history of three central Asian glaciers. This increases the value of the record from each glacier and enhances the likelihood that monitoring will continue. This region is at present a gap in our understanding of longer term glacier mass balance change around the globe that the WGMS reports. The study provides a detailed discussion of uncertainties and sensitivities that providing a useful example for other to adopt in a similar data limited environment. The use of snowline observations is a best practice for filling in gaps in a mass balance record as they represent direct observations of mass balance.

[Figure]

The comments below are numerous, but all are for minor additions or clarifications to enhance this valuable research.

2-8: signification to significance

2-15. Be consistent on spelling of Urumchi

3-3: Change Pelto (2011) to Pelto et al (2013)

Figure 3: Conveys important information, this could just as easily be conveyed in a table if there are production advantages to that.

5-14: Is the superimposed ice evidence indicative of persistent or transient existence?

7-7: Later in the paper it is worth simply mentioning the overall retreat observed on the three glaciers and how that fits into the negative balance regime.

8-18: Given that a couple of the references are related to the study I would add Mernild et al (2013) and Pelto et al (2013) where this is also discussed and are already references used elsewhere in paper.

8-25: Is the contrast with snow to firn weak for the terrestrial camera or just satellite images?

9-4: The initial position being the GPS location at the time of emplacement?

9-9: The extrapolation indicated is spatial. Given the field seasons occur before the end of the ablation season, is this also a temporal extrapolation model or is this different. Just clarify temporal from spatial extrapolations.

9-30: Given the issues described what is the vertical accuracy generally achieved? This is I think discussed at 13.24, but is appropriate here too.

10-8: Is the approach what Pelto (2010) and Mernild et al (2013) utilized which is TSL migration rate * balance gradient? This yields a directly observed ablation rate.

11-1: Why is the DDFs for Golubin very close to the maximum?

12-15: The approach and calibration methods are complex but appropriate. There is a simple validation that moves away from the model, that maybe what was used. That is what Pelto (2010) and Mernild et al (2013) utilized which is TSL migration rate * balance gradient. I do not suggest changing the approach, but this could be an independent validation for a few specific time periods on each glacier that is easy to apply, since both parameters are known.

13-6: Geodetic mass balance calculations do not account for internal accumulation either, unless it is incorporated in the density calculations, which typically does not occur.

13-29: Why this choice of 120 kgm-3, and what are implications vs a less conservative choice?

15-7: "These results demonstrate a relatively low sensitivity of the presented model to daily meteorological input data compared to mean seasonal data…

16-10: Section 4 was an excellent detailed summary fo the approach to determining errors and sensitivity. How did that lead to the error numbers here which are somewhat higher than I expected after seeing the details in Section 4.

18-9: The more negative balance years on Golubin and Abramov is where the TSL method generates more negative results. Could this be a reflection of melt low on glacier outside of ablation season that TSL does a better job of capturing? If this cannot be addressed to advantage than do not.

21-1: Could utilize Shea et al (2015) as well for support they found almost the same value for the Mount Everest region, different climate setting but still a high altitude monsoon influenced area. Wu et al (2011) also determine DDFs for Urumqi Glacier that could be referenced.

21-6: Why the large divergence for Abramov Glacier after 2009 between the constrained and unconstrained model?

22-15: The TSL observations also represent a direct point balance observations of considerable value.

23-9: Consult and refer to Bazhev (1986 or 1973) who directly measured the internal accumulation in firn on a glacier in the Pamirs and Abramov glacier. Found that almost all meltwater refroze in upper four layers of firn and the amount was in the 0.20 m/a range, this supports the approached used here. Further it is worth mentioning that such a study of internal accumulation should be redone some year as part of the mass balance program. Miller and Pelto (1999) observed a reduction in internal accumulation, on Lemon Creek Glacier, Alaska which if it occurs has impacts on the energy balance.

Figure 12: I do not think this is needed, if included redesign as the visual message is not well communicated by the approach used.

25-12: Worth noting that a SCAF time series such as in Figure 5 is also a critical value for water resource modelling, because of the different DDFs and DDFi values. Bazhev, A.: Infiltration and run-off of meltwater on glaciers. IASH Publ. 95 (Symposium at Cambridge 1969 – Hydrology of Glaciers), 245–250, 1973.

Bazhev, A.B.: Infiltration of meltwater on temperate and cold glaciers. Materialy Glyatsiologicheskikh Issedoveiy-Kronika. Obuzhdeniya, 58: 50-56, 1986.

Miller, M. M. and Pelto, M. S.: Mass Balance measurements on the Lemon Creek Glacier, Juneau Icefield, AK 1953–1998, Geogr. Ann., 81A, 671–681, 1999.

Pelto, M., Capps, D., Clague, J. J. and Pelto, B.: Rising ELA and expanding proglacial lakes indicate impending rapid retreat of Brady Glacier, Alaska. Hydrol. Process., 27: 3075–3082. doi:10.1002/hyp.9913, 2013.

Shea, J. M., Immerzeel, W. W., Wagnon, P., Vincent, C., and Bajracharya, S.: Modelling glacier change in the Everest region, Nepal Himalaya, The Cryosphere, 9, 1105-1128, https://doi.org/10.5194/tc-9-1105-2015, 2015.

Wu, L., Li, H. & Wang, L. J.: Application of a degree-day model for determination of mass balance of Urumqi Glacier No. 1, eastern Tianshan, China. Earth Sci. 22: 470. https://doi.org/10.1007/s12583-011-0201-x, 2011.

---

## Author Comment (AC1) · 26 Jan 2018

Below we respond to all comments by anonymous Reviewer 1. In response to this review, we changed the title of the manuscript and reworked the terminology. Additionally, we justified the independence of mass balances determined with in-situ point measurements with additional model experiments, completed and improved the description and calculation of geodetic mass change when snow-covered high-resolution images are used, and added statements concerning a broader applicability of the snowline approach.

[Figure]

The responses (normal font style) to the reviewers' comments (displayed in italic font style) are written directly into the reviews . The corresponding revised sentences in the manuscript are given in quotation marks.

We thank the referee for the valuable, constructive and detailed comments which improved the manuscript.

*(1) Title of the paper The title of the paper is misleading and needs to be changed. Indeed, in your approach the surface mass balance is modelled with meteorological data and this modelling is constrained by snowline observations during the ablation season. Thus, the title has to mention that the surface mass balance time series are quantified with a model. As it stands, one can have the impression that the surface mass balance is only inferred from snowline observations. I suggest: "Glacier surface mass balance modelling constrained by remote sensing derived snowlines. Application on three kyrgyz glaciers to quantify multi-decadal series". Or "The use of remote sensing derived snowlines to constrain glacier surface mass balance modeling. Application on ..."*

We agree that the title did not make reference to the modelling behind the approach, and adjusted it as suggested by the reviewer to "Multi-decadal mass balance series of three Kyrgyz glacier inferred from modelling constrained with repeated snowline observations."

———

*(2) Name of the proposed approach. In the same way, the name you give to the proposed approach (snowline-derived mass balance) is not adequate, and so is the*

*abbreviate in the tables and figures (Msnl). For the name of the approach, I propose: Modeled SMB constrained by snowline. And for the abbreviate: Mmod-tsl. TSL stands for Transient Snow Line and must be preferred to SNL.*

The name of the method was adjusted to "modelled SMB constrained by snowline observations" and we changed SNL to TSL throughout the entire manuscript. We also use "snowline approach" to refer to the method using a mass balance model constrained by snowline observations.

————

*(3) Terminology. "Mass balance", "glacier balance", etc: Be careful with the terminology and follow Cogley and others Glossary of mass balance terminology. For example, use "surface mass balance" instead of "mass balance" or "glacier balance", except when you refer to the geodetic mass balance.*

The terminology has been adjusted in the new version of the manuscript and we use surface mass balance wherever it is referred to. However, please note that when the modelled mass balance constrained by snowline observations is compared to geodetic mass balances, we do not refer to the surface mass balance but to a total mass change that includes an estimate of internal / basal mass balance.

————

*(4) Glaciological surface mass balance. The used method is not the classical glaciological one. Indeed, a model-based extrapolation is used and optimized to best fit the point measurements. What about the input data for accumulation and ablation (e.g. precipitation and temperature data)? Where do they come from? Are they the same as used in the model constrained by the snowline? If the data are the same, are the methods "constrained by the snowline observations" and "constrained by the point*

*measurements" strictly independent as they rely on the same meteorological data?*

This is an interesting and thoughtful question. Indeed, the same meteorological input is used for both approaches. However, we are convinced that both approaches can be considered as independent out of the following reasons:

The mass balance model used to derive the glaciological series is not regarded as a physical model, but as a statistical tool for obtaining glacier-wide surface mass balances based on field data and is closely tied to the field surveys. It is thus just a way to compute glacier-wide mass balance from stake measurements (such as the profile or the contour method). No model-based temporal extrapolation, e.g. to the hydrological year is involved. Consequentially, the climate data used in the model-based extrapolation of point mass balances permits spatial extrapolation to the entire glacier using physically-based equations and does not affect year-to-year variability which is given by the in-situ measurements. Here, we avoid the use of daily mass balance variability, and refer strictly to the surface mass balance obtained for the measurement dates.

In response to the reviewer's comment we performed a sensitivity test to prove the limited impact of the used meteorological time series on the glacier-wide mass balance computed from in-situ point measurements. We use artificially perturbed air temperature ($\pm 1 \circ$C) and precipitation ($\pm 25\%$) series. The standard deviation in the inferred glacier-wide surface mass balances is less than $0.01\,\mathrm{mm\,w.e.\,yr^{-1}}$ for all three glaciers and all years. This indicates that the method used to compute glaciological mass balance from in-situ field data exhibits a very small sensitivity on the actual meteorological data used for driving the mass balance model. The calculated annual glaciological SMB depends thus strongly on the in situ ablation and accumulation measurements and is considered as independent from the snowline approach. We tried to clarify the role of the meteorological data for the glaciological method used

here, and added the results of the sensitivity test.

"A model-based spatial extrapolation of point measurements to the entire glacier surface after Huss and others, (2009) was used to retrieve glacier-wide SMB for all years with direct measurements. The model is a combined distributed accumulation (Huss and others, 2008) and temperature index melt model with daily resolution (Hock, 1999) which was automatically optimized to best represent all collected point data from each seasonal/annual survey. The model is considered as a suitable tool to extrapolate the glaciological point measurements to the glacier surface for the measurement periods."

"For a sensitivity experiment we artificially shifted temperature and precipitation series used for the model-based extrapolation by $\pm 1 \circ$C and $\pm 25\%$, respectively. The resulting glacier-wide SMB indicate a very small sensitivity to the meteorological input data with a Standard Deviation (STD) of <0.01 mm w.e. yr$^{-1}$. We strictly refer to the annual SMB obtained for the measurement dates that are listed in table 2."

————

*(5) High resolution images recorded in November => impact of the snow cover on the derived DEM. Two of the stereo-pairs used to make DEMs and quantify the geodetic mass balance were recorded in November for Abramov and Golubin glaciers, respectively. I had a look at the SPOT catalogue and for Golubin I could find the SPOT7 images from 01/11/2014. Regarding Abramov, the images are not in the catalogue but I assume it is because these have been acquired within the SPIRIT acquisition campaign. Anyway, at least for 2014 and I assume it is the same for 2011, all the more because the SPOT5 images date from late November, the glaciers and surrounding terrains are completely snow covered. This implies several challenges for DEM generation:*
*- low contrast because of snow brightness*
*- unknown snow thickness*
*- impossibility to delineate the glacier outline.*
[Figure]

*How these issues have been tackled and what is their impact on the uncertainties? In addition, Table 7 shows that the geodetic mass balances for Abramov (2011-2015) and for Golubin (2006-2014) are less negative that the average annual surface mass balances quantified by your model constrained by the snowline. Can the snow-cover on the images implying a higher surface elevation (of an unknown value!) be a cause of this difference (or at least part of it)?*

This is an absolutely justified comment and we are aware that the quality of the two November scenes is not ideal. However, we are convinced to have extracted valuable information from the two images. Low contrast in the upper accumulation area due to fresh snow-cover and shading from steep mountain walls led to data gaps indicated as unmeasured areas on the glacier surface (26% for Abramov and 30% for Golubin, see also Figure 7 for Golubin). We accounted for the unmeasured areas on the glacier surface with a five-fold increase in uncertainty (see Section Uncertainties and Sensitivity). The retrieved information for the remaining glacier area appeared nevertheless reliable.

We agree with the reviewer regarding the issue of the unknown snow thickness. In principle, snow cover introduces a systematic bias into the calculation of geodetic volume change that needs to be accounted for. As it is unknown how deep the snow was on the glacier during the acquisition of the imagery we take a simple but efficient approach and account for the snow coverage by co-registering the digital elevation models using snow-covered stable ground sections. In this way, we approximately correct for the fresh snow on one of the scenes. This has been done for Abramov already in the first version of the submitted manuscript. However, for Golubin the vertical offset correction has now been adapted to systematically include snow-covered stable terrain sections. In addition, we compared the offset inferred using this approach to snow depth measurements at the Automatic Weather Station installed at 3300 m a.s.l. on the 01 of November 2014 and found good agreement. The geodetic mass

balance was corrected from $-0.22$ to $-0.30\,\mathrm{m\,w.e.\,yr^{-1}}$. which indeed shows a better agreement between the snowline-constrained model results and the geodetic mass balance for Golubin Glacier.

The outlines have been drawn on snow-free satellite images of other sensors as stated in Section "3.1 Glacier outlines". We did not include an error related to the delineation of the outlines in our error estimate.

In addition to the above correction, we have added some sentences about snow-cover conditions in the Study Site and Data section of the revised paper and have better described the approach to account for snow cover using the offset correction. Please note that especially for the geodetic mass balance of Golubin Glacier, it was difficult to find appropriate high-resolution images. The estimated uncertainties are considerably higher when snow-covered images were used than for the other geodetic estimates.

"Important snow coverage was present on the SPOT5 image from 2011 for Abramov and on the SPOT7 image from 2014 for Golubin. A fine layer of fresh snow covered parts of the SPOT6 image from 2015 for Glacier No. 354."

"For this vertical co-registration, only terrain sections with a slope lower than approximately 30° were selected and areas with parallax-matching problems were avoided. Snow-covered areas were included in the offset correction, in order to correct for fresh snow."

"The geodetic method reveals a total mass balance of $-0.30 \pm 0.42\,\mathrm{m\,w.e.\,yr^{-1}}$ for Golubin Glacier from 8 September 2006 to 1 November 2014 (Fig. 10). The corresponding total modelled mass balance constrained by snowline observations was slightly more negative with $-0.38 \pm 0.35\,\mathrm{m\,w.e.\,yr^{-1}}$ for the same period (Table 8 and Fig. 11)."

————

*(6) Surface mass balance interannual variability. It is pity that your discussion about the interannual variability of the SMB is short and only dedicated on the annual values. You should have a look on the two terms of the annual surface mass balance, and discuss about their interannual variability. You could see if the interannual variabilities of accumulation and ablation are homogeneous, comparable between the three sites and if the difference you mention regarding the interannual variability of Glacier No. 354 annual SMB is more likely related to a different ablation or accumulation interannual variability.*

We agree with the reviewer in general. An analysis of the interannual variability would be very interesting and insightful. However, the data sets available in this study (annual in-situ point mass balance, decadal geodetic mass balance, repeated snowline observations) only allow us to directly constrain annual mass balances. Seasonal and daily mass balances, calculated by the snowline-constrained model are not tied directly to any observations and day-to-day variability depends on partly uncertain meteorological information. Therefore they are subject to larger unknown uncertainties. Furthermore, the calibration of the two parameters $DDF_{\text{snow}}$ and $C_{\text{prec}}$ are not strictly independent and thus could lead to important under- or overestimations of the two seasonal components of the modelled surface mass balances. This shortcoming of the method is described in Section Discussion and shown in Figure 13b. Unfortunately, at the current stage, we only have a few seasonal in situ measurements for the studied glaciers. A reliable verification of the performance of the snowline approach in terms of the winter and summer surface mass balance is thus not possible. For the above-mentioned reasons we avoid an interpretation of the interannual variability here.

————

*(7) Application of the proposed approach. You mention that the approach you propose can be useful to quantify surface mass balance time series for a number of remote glaciers. Although I mostly agree with this statement, I wonder if the approach can*

*indeed be applicable for summer accumulation glaciers (like in the tropics, or in monsoon regime regions) or for high latitudes glaciers where superimposed ice can be more important than in Kyrgyzstan. You can probably add some sentences on this point in the conclusion.*

We added some sentences on the potential wider applicability of the proposed approach in the Discussion section of our manuscript. However, we are unable to perform a complete assessment of the transferability of our approach to other climate zones in the scope of the present study.

"SMBs inferred from modelling constrained with repeated snowline observations is closely tied to the TSL observations. The method might be able to yield reliable surface mass balance estimates for many glaciers in different climatic regimes, for which the transient snowline is an indicator of the surface mass balance. The relationship between the snowline and the surface mass balance can however be importantly challenged when the position of the transient snowline is blurred by fresh snow or superimposed ice. The applicability of the snowline approach presented here, can thus be critical when the transient snowline on remote sensing data cannot unambiguously be identified. This is mainly a problem for glaciers with a summer accumulation regime due to frequent fresh snow, and glaciers with a high relevance of superimposed ice."

————

Specific comments:

Abstract
*P1, L5-6: the sentence "A combination of 3 independent [. . .] Golubin and No. 354"*
*needs to be reformulated. Indeed, the methods are not combined to reconstruct the*
*surface mass balance. The methods are compared/cross-validated but not combined.*
*For me, you would have a combination if, for example, your modeled surface mass*
*balance time series had been adjusted with the geodetic method.*

Reformulated.
"By cross-validating the results of three independent methods, we reconstructed the mass balance of the
three benchmark glaciers, Abramov, Golubin and No. 354 for the past two decades."
————

*P1, L8-9: "satellite optical imagery" instead of "satellite imagery".*

Done.
————

*P1, L12, 13 and in the entire paper: should write "yr-1", instead of "a-1". "yr-1" is most*
*common except for IGS journals.*

Done.
————

*P1, L15: prefer "unmonitored" to "inaccessible". I do not know about an inaccessible*

*glacier on Earth."*

Done.

———

1. Introduction
*P2, L2: should the IPCC reference be quoted Stoecker and others, 2013?*

Referencing has been changed.

———

*P2, L18: remove a comma after "e.g.,"*

Done.

———

2. Study site and data
*P4, L12-13: this statement is useless if you do not give any quantification.*

Quantification has been added.
"For Golubin, the mass loss reported by Bolch, (2015) was $-0.28 \pm 0.96$ m w.e. yr$^{-1}$ from 2000 to 2012, whereas Brun and others (2017) found a mass balance of $-0.04 \pm 0.19$ m w.e. yr$^{-1}$ for the period 2002 to 2013."

———

*P5, L11-12: idem, give a value.*

Quantification has been added.

"Mass loss since the mid-1970s was reported by different studies (Pieczonka and others, 2015, Kronenberg and others, 2016, Brun and others, 2017) ranging from about $-0.8$ to $-0.5\,\mathrm{m\,w.e.\,yr^{-1}}$."

————————

*P5, L6 and 15: should mention the elevation of the AWS*

Done.

————————

*Figure 3: for Glacier No. 354, the Quickbird and Pléiades images recorded in 2003 and 2015 respectively have not been used for snowline mapping? If yes, these images should appear in the Figure. If not, why?*

The QuickBird image has not been used for snowline mapping because it dates from summer 2003, and the first year to compute was the year 2004. We clarified in the caption and manuscript text.

"Image availability and distribution for snowline mapping. Numbers indicate the total available scenes per year and glacier. Prior to 1998, image coverage is sparse for all three glaciers. For Golubin and for Glacier No. 354, the first summer season for which enough snowline observations could be collected was 2000 and 2004, respectively. Snow-covered high-resolution images have not been used to delineate the snowline and are not shown here."

————————

*Table 1: should write "snowline" instead of SNL. You should also indicate the sensor in brackets for the high resolution images. If I am correct these are SPOT5 and Pléiades for Abramov, ALOS and SPOT7 for Golubin and Quickbird, GeoEye and Pléiades for*

*No. 354.*

Done.

————

3. Methods
*See my main comment related to this section (no4), and the one regarding the terminology for the title of this sub-section.*

Please refer to the answer given for the main comment No. 4.

————

*P10, L23: the paper by Huintjes and others has not been finally published. I wonder if papers that stayed in discussion and/or were rejected can be quoted. You can probably remove this reference from the text and the ref list, all the more that it is quoted within a list of 5 references starting by e.g.*

The reference has been removed.

————

*In Table 2 and in the text (e.g. P11, L7): I recommend using 'Z' instead of 'H' for the elevation criteria. H usually stands for thickness.*

Done.

————

4. Uncertainties and model sensitivity

*P15, L3-8: you should provide an illustration for the test that uses average daily temperature and precipitation series. In addition, you have to explain the low sensitivity of your model to the input meteorological data. I assume this is because the parameters are adjusted for each year and for each glacier.*

An illustration has been added and we have extended our explanation on the low sensitivity of the model to the meteorological input data (see attachement fig. 1)

"These results demonstrate a relatively low sensitivity of our model approach to daily meteorological input data. With the chosen calibration procedure the model parameters $DDF$snow and $C_{\mathrm{prec}}$ are adjusted to best represent the TSL observations for each year and glacier individually. The modelled SMB are thus closely tied to the snowline observations and exhibit a reduced dependence from meteorological input data."

————

*Figure 5b: change JAJ by JJA*

Done.

————

5. Results
*P16, L3-6: you should provide an illustration for the comparison between SCAFs given by the model and the images for all the used images, glacier by glacier.*

We added an illustration comparing the observed and modelled SCAF for each glacier (see attachement: fig. 2).

————

*P16, L7-8: the period 1998 to 2016 stands for Abramov Glacier only. You have to*

*mention the specific periods for each glacier. Golubin starts in 2000 and No. 354
in 2004. In addition "over the two last decades" can be removed from the sentence
because the time periods for each glacier will be mentioned.*

Done.

"Annual glacier-wide modelled surface mass balances constrained by snowline observations, calculated
for Abramov (1998-2016), for Golubin (2000-2016), and for Glacier No. 354 (2004-2016), located in the
Pamir-Alay and the Tien Shan, are predominantly negative (Fig. 8 and Tabel 6)".

————

*P16, L8: you refer to Figure 8, but because figures 6 and 7 have not been quoted yet,
this figure should be Figure 6. Anyway, because I suggest adding two more figures, it
will probably remain Figure 8, but must appear before the current figures 6 and 7.*

Numbering has been corrected.

————

*Table 5: you must indicate in the table itself (not only the caption) on which period the
STD is quantified*

Done.

————

*P17, L1: should the first close-to-zero SMB period be extended to 2005?*

We agree and extended the period to 2005.

————

*P17, L1: "Glacier No. 354, situated in a more continental climate regime, [. . .]". I am a bit skeptical with this statement! All three glaciers are in a continental regime. This glacier being located in the inner range, it might receive less precipitation than the two others. Is it what you mean? See also my main comment related to interannual variability of the SMB (no6).*

Indeed, we refer to a strong precipitation gradient from West to East for both the Pamir/Pamir-Alay and the Tien Shan and also a different precipitation distribution. Glaciers in the western part of the Tien Shan receive more winter accumulation whereas glaciers located more in the east are subject to considerable summer ac-cumulation. This corresponds to the general synoptic large-scale meteorological con-ditions over Central Asia, influenced by the main direction of the zonal flow of the air masses from west to east. According to Balashova and others (1960, Hydrometeoro-logical Publishing House: Leningrad) and Schienmann and others (2008, International Journal of Climatology, vol. 28(3)) also meridional air mass flow can occur. This occurs either in situations when tropical air masses enter from South and south-west or when north-westerly, northerly and sometimes even north-easterly, cold air masses intrude into Central Asia. We specified the statement but do not want go into more detail in the manuscript.

"Glacier No. 354, receiving lower amounts of total annual precipitation than the other two glaciers studied here, shows the weakest interannual variability and has a positive balance only in 2009 (Table 6)."

————

*P17, L4-5: this sentence refers to the years 2006 and 2008 mentioned in the previous sentence? If yes, the two sentences might be separated by a semi-colon not a dot.*

Done.

————

*P17, L8-9: same thing here, the two sentences could be separated by a semi-colon not a dot.*

Done.

————

*Table 6: you should mention in the caption that the values differ from Table 5 because the glacier-wide annual surface mass balances are not computed over the same number of days. However, the difference is really high for some years, for example 2014 for Golubin Glacier (more than 0.8 m w.e. different!). You could indicate the number of days differing from the quantification given in Table 5.*

Instead of indicating the number of differing days, we added a table with the exact survey dates, and changed the statement in the caption to underline the difference to Table 5.

"Annual SMB $B_{\mathrm{sfc(meas)}}$ for the measurement periods (i.e., exact dates of the surveys, Table 2) based on direct glaciological surveys and on the snowline-constrained model for the three glaciers in m w.e yr$^{-1}$."

————

*Table 7. A dot is missing after 'e' in "m w.e a-1" In addition, regarding Abramov Glacier, why the first period is 2003-2015 and not 2003-2011?*

Due to the limited image quality of the SPOT image from November 2011, we considered the mass balance from 2003 to 2015 to be more robust, and decided to show the result for this period instead.

————

6. Discussion

*P20, L3: replace "integrating" by "using"*

Done.

————

*P21, L6: discuss why the difference is opposite for Glacier No. 354*

We discovered an error in the model settings of Glacier No. 354 of the unconstrained model leading to the too positive balance for Glacier No. 354. The error is corrected and we would like to apologize for this mistake. The calculations and figures are now updated (see attachement Fig. 3). Not surprisingly, the mass balance is much more negative for the unconstrained run.

————

*P22, L27: replace "too positive" by "not negative enough"*

Done.

————

*P22, L33: change "shows" by "showed"*

Done.

————

*P23, L18-20: you indicate that the average glacier-wide annual surface mass balance quantified by Brun and others (2017) shows a stronger mass loss than your study. This is not really exact. The difference is important with your modeling approach, but*

*the estimate by Brun and others and your geodetic estimate are really close.*

We clarified our statement.

"The average mass balance for Abramov of $-0.38 \pm 0.10\,\text{m w.e. yr}^{-1}$ (2002-2014) derived by Brun and others, (2017) using multi-temporal ASTER DEMs indicates a stronger mass loss than the results obtained with the snowline approach. We note, however, that the start and end dates of their geodetic mass balance represent a mean over a mosaic of different dates, thus hampering direct comparison. In addition, the differences are still within their error bounds. The results by Brun and others, (2017 are, however, in line with the geodetic mass balance calculated in this study for the period 2003 to 2015 based on high-resolution satellite images."

————

————————————————————

[Figure]

[Figure]

**Fig. 1.** Comparison between the annual surface mass balance obtained from the snowline-constrained model when using meteorological and climatological average daily data for Abramov (squares), Golubin (diamonds

[Figure]

Fig. 2. Comparison between the observed SCAF and the modelled SCAF for for Abramov (squares), Golubin (circles), and Glacier No. 354 (diamonds).
**Fig. 3.** Comparison of the cumulative surface mass balance derived from unconstrained mass balance modelling to the results obtained from snowline constrained modelling from 2004 to 2016.

---

## Author Comment (AC2) · 26 Jan 2018

Below we respond to all comments by Prof. Dr. M. Pelto (Reviewer 2). We mainly tried to clarify and complete imprecise sections, to remove inconsistencies and added suggested references in the revised manuscript.

The responses (normal font style) to the reviewers' comments (displayed in italic font style) are written directly into the reviews . The corresponding revised sentences in the manuscript are given in quotation marks.

[Figure]

We thank the referee for the valuable, constructive and detailed reflections and comments. The comments certainly improve the manuscript.

*2-8: signification to significance*

Done.

———

*2-15. Be consistent on spelling of Urumchi*

This has been corrected through the entire manuscript. We use Urumqi.

———

*3-3: Change Pelto (2011) to Pelto et al (2013)*

Done.

———

*Figure 3: Conveys important information, this could just as easily be conveyed in a table if there are production advantages to that*

We prefer to illustrate the image availability in a figure rather than in a table. The individual image dates are not very important in this context. However, we would like to sketch the increased image availability with time which, in our opinion, becomes more

evident with a figure.

———

*5-14: Is the superimposed ice evidence indicative of persistent or transient existence?*

The superimposed ice zone is rather persistent. We clarified this in the manuscript.
"Evidence of persistent superimposed ice is found and Kronenberg and others (2016) estimated internal accumulation to be $+0.04\,\mathrm{m\,w.e.\,yr^{-1}}$."

———

*7-7: Later in the paper it is worth simply mentioning the overall retreat observed on the three glaciers and how that fits into the negative balance regime.*

We added some statements on the retreat pattern of the glaciers and brought it into context with the negative balance regime in the Discussion section.
"For Golubin and Glacier No. 354 a slightly more negative annual average balance of $-0.41 \pm 0.33\,\mathrm{m\,w.e.\,yr^{-1}}$ and $-0.36 \pm 0.32\,\mathrm{m\,w.e.\,yr^{-1}}$, respectively, was calculated for the same time period (Table 6). Length change measurements underline the observed negative balance regime of all three glaciers (Hoelzle and others, 2017). A predominant glacier retreat was observed for the last century. A first speed-up of frontal retreat occurred in the 1980s and acceleration was observed in the last decade. However, no clear acceleration of mass loss for the three glaciers was identified over the investigated periods. Two phases of close-to-zero SMB could be recognized (2002-2005 and 2009-2011) for all glaciers."

———

*8-18: Given that a couple of the references are related to the study I would add Mernild et al (2013) and Pelto et al (2013) where this is also discussed and are already references used elsewhere in paper.*

Done.
———

*8-25: Is the contrast with snow to firn weak for the terrestrial camera or just satellite images?*

Clarified.
"Contrast becomes rather weak, especially when the snowline rises above the firn line (Rabatel and others, 2013, Want and others, 2014) for both satellite and terrestrial camera images."
———

*9-4: The initial position being the GPS location at the time of emplacement?*

The GPS measurements of the glacier front position were repeated every year.
"Annually repeated measurements of the glacier front position using a handheld GPS for all three glaciers were combined with the satellite observations for mapping from 2011 onwards."
———

*9-9: The extrapolation indicated is spatial. Given the field seasons occur before the end of the ablation season, is this also a temporal extrapolation model or is this different. Just clarify temporal from spatial extrapolations.*

Clarified in the manuscript. The extrapolation is only spatial. We did not adjust the glaciological mass balance for the start and end date of the hydrological year, but calculated the mass balance derived with the snowline approach to match the dates of the direct measurements for comparison.
"A model-based spatial extrapolation of point measurements to the entire glacier surface after Huss and

others, (2009) was used to retrieve glacier-wide SMB for all years with direct measurements. The model is a combined distributed accumulation (Huss and others, 2008) and temperature index melt model with daily resolution (Hock, 1999) which was automatically optimized to best represent all collected point data from each seasonal/annual survey. The model is considered as a suitable tool to extrapolate the glaciological point measurements to the glacier surface for the measurement periods."
————

*9-30: Given the issues described what is the vertical accuracy generally achieved? This is I think discussed at 13.24, but is appropriate here too.*

We added the vertical accuracy.
"The vertical offset was reduced to 1.0 m (2003-2015) and to 0.6 m (2011-2015) for Abramov, to 0.7 m for Glacier No. 354 and to 1.8 m for Golubin."
————

*10-8: Is the approach what Pelto (2010) and Mernild et al (2013) utilized which is TSL migration rate \* balance gradient? This yields a directly observed ablation rate.*

No, we did not use the same approach as presented in Pelto (2010) and Mernild and others (2013). These approaches integrate direct field measurements to calculate snow ablation rates which are not consistently available for our study region. Here, we use an iterative modelling approach to calculate snow accumulation and use only TSL as input. Please refer to Section 4.3 that gives details on the model constrained by snowline observations.
————

*11-1: Why is the DDFs for Golubin very close to the maximum?*

Thanks for spotting this. It was an error in the submitted manuscript. The mean DDF is $5.09\,\mathrm{mm\,day^{-1}\,{}^{\circ}C^{-1}}$ and not $5.49\,\mathrm{mm\,day^{-1}\,{}^{\circ}C^{-1}}$. We apologize for this mistake.

————

*13-6: Geodetic mass balance calculations do not account for internal accumulation either, unless it is incorporated in the density calculations, which typically does not occur.*

This is a very interesting comment on the problem of geodetic mass balance computations. We are however unable to fully resolve this issue in the present paper. We completely agree with the reviewer that the density assumption to calculate the geodetic mass balance is critical and so far not well understood. In principle, the geodetic mass balance, however, includes all mass changes within a glacier, and not only surface processes. The problem goes back to the density assumptions of converting volume to mass changes.

We are sure that a correction is needed for unifying surface mass balances with geodetic surveys that monitor all mass change components. However, as stated by the reviewer, to correctly account for internal accumulation within the geodetic mass balance, a correct density assumption is required. Yet, this is not straight-forward, and we believe that improving the confidence in volume-to-mass change density assumptions is not possible within the scope of this study and more process-related studies on this subject are required. We are aware that chosen density assumption is a strong simplification but at the current stage, we are simply not able to reasonably correct the calculated geodetic mass balance for the component of internal accumulation. This is why we have chosen an error ranges for the volume-to-mass conversion which are expected to cover the respective uncertainties.

————

*13-29: Why this choice of 120 kgm-3, and what are implications vs a less conservative choice?*

As mentioned above, the density assumption is very critical to convert volume to mass change. Currently, we have unfortunately not enough knowledge to make more adequate assumptions. Nevertheless, we decided to use a more conservative volume-to-mass conversion and simply doubled the uncertainty range of the density for periods shorter than 3 years.

————

*15-7: "These results demonstrate a relatively low sensitivity of the presented model to daily meteorological input data compared to mean seasonal data...".*

The sentence has been changed according to the comments of reviewer 1.

"These results demonstrate a relatively low sensitivity of our model approach to daily meteorological input data. With the chosen calibration procedure the model parameters $DDF$snow and $C_{\text{prec}}$ are adjusted to best represent the TSL observations for each year and glacier individually. The modelled SMB are thus closely tied to the snowline observations and exhibit a reduced dependence from meteorological input data."

————

*16-10: Section 4 was an excellent detailed summary of the approach to determining errors and sensitivity. How did that lead to the error numbers here which are somewhat higher than I expected after seeing the details in Section 4.*

We combined the different error sources for each year by the Root Sum of Squares, assuming independence between the different error components and averaged the annual errors for the considered periods. We clarified this in the manuscript.

"Components 1 to 5 are assumed to be independent of each other and are combined as RSS to represent the total error of the annual SMB $\sigma_{\mathrm{tsl}}$ obtained from the snowline approach. We then averaged the annual error over the different periods to compute the uncertainty for the respective periods."

———————

*18-9: The more negative balance years on Golubin and Abramov is where the TSL method generates more negative results. Could this be a reflection of melt low on glacier outside of ablation season that TSL does a better job of capturing? If this cannot be addressed to advantage than do not.*

As the daily mass balance evolution is a pure product of our modelling approach and per se not actually constrained by the available observations (in contrast to the annual mass balance), we prefer to leave aside interpretations on the seasonal components of the surface mass balance (see also comment to reviewer 1 above).

———————

*21-1: Could utilize Shea et al (2015) as well for support they found almost the same value for the Mount Everest region, different climate setting but still a high altitude monsoon influenced area. Wu et al (2011) also determine DDFs for Urumqi Glacier that could be referenced*

We integrated the suggested references to underline the choice of our parameters.

"We chose $C_{\mathrm{prec}}$ to account for a 20%-measurement error of the recorded precipitation (Sevruk, 1981), and a combination for $DDF_{\mathrm{ice}}$ (7.0 mm day$^{-1}$ °C$^{-1}$) and $DDF_{\mathrm{snow}}$ (5.5 mm day$^{-1}$ °C$^{-1}$) as recommended by Hock, (2003) for the former Soviet territory, for all three glaciers. Similar values were used to model glaciers in the Tien Shan and Himalayas (e.g., Zhang and others, 2006, Wu and others, 2011, Shea and others, 2015)."

———————

*21-6: Why the large divergence for Abramov Glacier after 2009 between the con-strained and unconstrained model?*

2009 was a rather cold and snow-rich year. Especially summer snowfall reduced the melt for most glaciers in the region (Barandun and others, 2015, Kronenberg and others, 2016, Kenzhebaev and others, 2017). A less negative mass balance can be observed for all three glaciers in the results obtained from both models.

The divergence between the constrained and unconstrained model increases after 2011. SMB are much more negative for the unconstrained model from 2011 to 2016 than before. This is most likely due to the change of the meteorological input datasets. In 2011, the new meteorological station was installed, and we replaced the Reanalysis temperature data with measured air temperature. With the use of the snowline data the change of the meteorological data is secondary because of the decreased sensitivity towards the meteorological input. However, when using an unconstrained model, such effects can be quite large as shown in Figure 12.

———————

*22-15: The TSL observations also represent a direct point balance observations of considerable value.*

We agree with the reviewer. However, the use of the transient snowline as point mass balance observation is not the focus of this paper and we prefer not to provide more details on this subject here in order to keep our article focussed.

———————

*23-9: Consult and refer to Bazhev (1986 or 1973) who directly measured the internal accumulation in firn on a glacier in the Pamirs and Abramov glacier. Found that*

*almost all meltwater refroze in upper four layers of firn and the amount was in the 0.20 m/a range, this supports the approached used here. Further it is worth mentioning that such a study of internal accumulation should be redone some year as part of the mass balance program. Miller and Pelto (1999) observed a reduction in internal accumulation, on Lemon Creek Glacier, Alaska which if it occurs has impacts on the energy balance.*

For the estimate by Barandun and others (2015) from where we adopted the estimate of internal accumulation, the study by Bazhev (1973, IASH Publ. 95). was used for calibrating the refreezing model. In Barandun and others (2015), the quantification of refreezing is based on calculating a temperature profile in firn and ice using the heat conduction equation (see, e.g., Pfeffer and others, 1991, Journal Geophysical Research, 96). The refreezing model is calibrated by adjusting firn temperature at the bottom of the profile at model initialization to match repeated firn temperature measurements made in three firn cores (Glazirin and others, 1993). The results were finally compared to findings for Abramov presented by Bazhev (1973, IASH Publ. 95).

During recent field visits, we measured firn temperature at two locations in the accumulation zone. No negative temperatures during summer field campaign where found indicating that all energy available for refreezing melt water had been used. A new project (since April 2017) aims at analysing the firn stratigraphy, and at quantifying of refreezing on Abramov Glacier. However, no results are available yet. We thus decided not to add more information on the calculated internal balance but refer to published former work.
———

*Figure 12: I do not think this is needed, if included redesign as the visual message is not well communicated by the approach used.*

We would like to keep the figure in the manuscript. We are nevertheless very open for suggestions to better communicate the visual message.
————

*25-12: Worth noting that a SCAF time series such as in Figure 5 is also a critical value for water resource modelling, because of the different DDFs and DDFi values.*

We agree that the method carries a large potential to retrieve mass balance information at higher temporal scale that might be useful for water resource management. However at the moment, the uncertainties related to the daily mass balance series are high and we prefer to refer only to the annual balance components (see also comments above).
————

————————————————————

---

## Author Response (AR1)

**Coverletter**

**Multi-decadal mass balance series of three Kyrgyz glaciers inferred from modelling constrained with repeated snowline observations.**

Martina Barandun, Matthias Huss, Ryskul Usubaliev, Erlan Azisov, Etienne Berthier, Andreas Kääb, Tobias Bolch and Martin Hoelzle

Dear Editor,

The two constructive reviews were very helpful to finalize the paper. In particular, we have
- adjusted the title and reworked the terminology,
- justified the independence of mass balances determined with in-situ point measurements with additional model experiments,
- completed and improved the description and calculation of geodetic mass change and
- added statements concerning a broader applicability of the snowline approach

Below, we respond to all comments, and state how we account for them in the revised version of the paper. The responses (normal font style) to the reviewers' comments are written directly into the reviews (displayed in italic font style). The corresponding revised sentences in the manuscript are given in quotation marks, including page and line numbers in the revised manuscript.

**Comments of Reviewer #1 (Anonymous)**

*(1) Title of the paper The title of the paper is misleading and needs to be changed. Indeed, in your approach the surface mass balance is modelled with meteorological data and this modelling is constrained by snowline observations during the ablation season. Thus, the title has to mention that the surface mass balance time series are quantified with a model. As it stands, one can have the impression that the surface mass balance is only inferred from snowline observations. I suggest: "Glacier surface mass balance modelling constrained by remote sensing derived snowlines. Application on three kyrgyz glaciers to quantify multi-decadal series". Or "The use of remote sensing derived snowlines to constrain glacier surface mass balance modeling. Application on ..."*

We agree that the title did not make reference to the modelling behind the approach, and adjusted it as suggested by the reviewer to "Multi-decadal mass balance series of three Kyrgyz glacier inferred from modelling constrained with repeated snowline observations."

———

*(2) Name of the proposed approach. In the same way, the name you give to the proposed approach (snowline-derived mass balance) is not adequate, and so is the abbreviate in the tables and figures (Msnl). For the name of the approach, I propose: Modeled SMB constrained by snowline. And*

*for the abbreviate: Mmod-tsl. TSL stands for Transient Snow Line and must be preferred to SNL.*

The name of the method was adjusted to "modelled SMB constrained by snowline observations"and we changed SNL to TSL throughout the entire manuscript. We also use "snowline approach"to refer to the method using a mass balance model constrained by snowline observations.

————

*(3) Terminology. "Mass balance", "glacier balance", etc: Be careful with the terminology and follow Cogley and others Glossary of mass balance terminology. For example, use "surface mass balance"instead of "mass balance"or "glacier balance", except when you refer to the geodetic mass balance.*

The terminology has been adjusted in the new version of the manuscript and we use surface mass balance wherever it is referred to. However, please note that when the modelled mass balance constrained by snowline observations is compared to geodetic mass balances, we do not refer to the surface mass balance but to a total mass change that includes an estimate of internal / basal mass balance.

————

*(4) Glaciological surface mass balance. The used method is not the classical glaciological one. Indeed, a model-based extrapolation is used and optimized to best fit the point measurements. What about the input data for accumulation and ablation (e.g. precipitation and temperature data)? Where do they come from? Are they the same as used in the model constrained by the snowline? If the data are the same, are the methods "constrained by the snowline observations" and "constrained by the point measurements"strictly independent as they rely on the same meteorological data?*

This is an interesting and thoughtful question. Indeed, the same meteorological input is used for both approaches. However, we are convinced that both approaches can be considered as independent out of the following reasons:

The mass balance model used to derive the glaciological series is not regarded as a physical model, but as a statistical tool for obtaining glacier-wide surface mass balances based on field data and is closely tied to the field surveys. It is thus just a way to compute glacier-wide mass balance from stake measurements (such as the profile or the contour method). No model-based temporal extrapolation, e.g. to the hydrological year is involved. Consequentially, the climate data used in the model-based extrapolation of point mass balances permits spatial extrapolation to the entire glacier using physically-based equations and does not affect year-to-year variability which is given by the in-situ measurements. Here, we avoid the use of daily mass balance variability, and refer strictly to the surface mass balance obtained for the measurement dates.

In response to the reviewers comment we performed a sensitivity test to prove the limited impact of the used meteorological time series on the glacier-wide mass balance computed from in-situ point measurements. We use artificially perturbed air temperature ($\pm 1°$C) and precipitation ($\pm 25\%$) series. The standard deviation in the inferred glacier-wide surface mass balances is less than $0.01\,\mathrm{mm\,w.e.\,yr^{-1}}$ for all three glaciers and all years. This indicates that the method used

to compute glaciological mass balance from in-situ field data exhibits a very small sensitivity on the actual meteorological data used for driving the mass balance model. The calculated annual glaciological SMB depends thus strongly on the in situ ablation and accumulation measurements and is considered as independent from the snowline approach. We tried to clarify the role of the meteorological data for the glaciological method used here, and added the results of the sensitivity test.

**Page 9, Lines 10-15**

"A model-based spatial extrapolation of point measurements to the entire glacier surface after Huss et al. (2009) was used to retrieve glacier-wide SMB for all years with direct measurements. The model is a combined distributed accumulation (Huss et al., 2008) and temperature-index melt model with daily resolution (Hock, 1999) which was automatically optimized to best represent all collected point data from each seasonal/annual survey. The model is considered as a suitable tool to extrapolate the glaciological point measurements to the glacier surface for the measurement periods."

**Page 14, Lines 6-9**

"For a sensitivity experiment we artificially shifted temperature and precipitation series used for the model-based extrapolation by $\pm 1°$C and $\pm 25\%$, respectively. The resulting glacier-wide SMB indicate a very small sensitivity to the meteorological input data with a Standard Deviation (STD) of $<0.01\,\mathrm{mm\,w.e.\,yr^{-1}}$. We strictly refer to the annual SMB obtained for the measurement dates that are listed in table 2."

————

*(5) High resolution images recorded in November - impact of the snow cover on the derived DEM. Two of the stereo-pairs used to make DEMs and quantify the geodetic mass balance were recorded in November for Abramov and Golubin glaciers, respectively. I had a look at the SPOT catalogue and for Golubin I could find the SPOT7 images from 01/11/2014. Regarding Abramov, the images are not in the catalogue but I assume it is because these have been acquired within the SPIRIT acquisition campaign. Anyway, at least for 2014 and I assume it is the same for 2011, all the more because the SPOT5 images date from late November, the glaciers and surrounding terrains are completely snow covered. This implies several challenges for DEM generation:*
*- low contrast because of snow brightness*
*- unknown snow thickness*
*- impossibility to delineate the glacier outline.*
*How these issues have been tackled and what is their impact on the uncertainties? In addition, Table 7 shows that the geodetic mass balances for Abramov (2011-2015) and for Golubin (2006-2014) are less negative that the average annual surface mass balances quantified by your model constrained by the snowline. Can the snow-cover on the images implying a higher surface elevation (of an unknown value!) be a cause of this difference (or at least part of it)?*

This is an absolutely justified comment and we are aware that the quality of the two November scenes is not ideal. However, we are convinced to have extracted valuable information from the two images. Low contrast in the upper accumulation area due to fresh snow-cover and shading from steep mountain walls led to data gaps indicated as unmeasured areas on the glacier surface (26% for Abramov and 30% for Golubin, see also Figure 7 for Golubin). We accounted for the unmeasured areas on the glacier surface with a five-fold increase in uncertainty (see Section "Uncertainties and Sensitivity"). The retrieved information for the remaining glacier area appeared nevertheless reliable. Also, the problems with low visual contrast are much reduced for

modern sensors (here: 12 bit) and careful gain management, compared to older 8 bit sensors that often showed saturation over snow. In essence, in our cases there was over large areas sufficient contrast for parallax matching even over snow.

We agree with the reviewer regarding the issue of the unknown snow thickness. In principle, snow cover introduces a systematic bias into the calculation of geodetic volume change that needs to be accounted for. As it is unknown how deep the snow was on the glacier during the acquisition of the imagery we take a simple but efficient approach and account for the snow coverage by co-registering the digital elevation models using snow-covered stable ground sections. In this way, we approximately correct for the fresh snow on one of the scenes. This has been done for Abramov already in the first version of the submitted manuscript. However, for Golubin the vertical offset correction has now been adapted to systematically include snow-covered terrain sections. In addition, we compared the offset inferred using this approach to snow depth measurements at the Automatic Weather Station installed at 3300 m a.s.l. on the 1 of November 2014 and found good agreement. The geodetic mass balance was corrected from $-0.22$ to $-0.30 \, \mathrm{m \, w.e. \, yr^{-1}}$. which indeed shows a better agreement between the snowline-constrained model results and the geodetic mass balance for Golubin Glacier.

The outlines have been drawn on snow-free satellite images of other sensors as stated in Section "Glacier outlines". We did not include an error related to the delineation of the outlines in our error estimate.

In addition to the above correction, we have added some sentences about snow-cover conditions in the Study Site and Data section of the revised paper and have better described the approach to account for snow cover using the offset correction. Please note that especially for the geodetic mass balance of Golubin Glacier, it was difficult to find appropriate high-resolution images. The estimated uncertainties are considerably higher when snow-covered images were used than for the other geodetic estimates.

**Page 5, Lines 27-29**
"Important snow coverage was present on the SPOT5 image from 2011 for Abramov and on the SPOT7 image from 2014 for Golubin. A fine layer of fresh snow covered parts of the SPOT6 image from 2015 for Glacier No. 354."

**Page 10, Lines 4-6**
"For this vertical co-registration, only terrain sections with a slope smaller than $30\,^\circ$ were selected and areas with parallax-matching problems were avoided. Snow-covered areas were included in the offset correction, in order to correct for fresh snow on the image, assuming similar snow thicknesses on- and off-glacier."

**Page 19, Lines 17-19**
"The geodetic method reveals a total mass balance of $-0.30 \pm 0.37 \, \mathrm{m \, w.e. \, yr^{-1}}$ for Golubin Glacier from 8 September 2006 to 1 November 2014 (Fig. 10). The corresponding total modelled mass balance constrained by snowline observations was slightly more negative with $-0.38 \pm 0.35 \, \mathrm{m \, w.e. \, yr^{-1}}$ for the same period (Table 8 and Fig. 11)."
————

*(6) Surface mass balance interannual variability. It is pity that your discussion about the interannual variability of the SMB is short and only dedicated on the annual values. You should*

*have a look on the two terms of the annual surface mass balance, and discuss about their inter-annual variability. You could see if the interannual variabilities of accumulation and ablation are homogeneous, comparable between the three sites and if the difference you mention regarding the interannual variability of Glacier No. 354 annual SMB is more likely related to a different ablation or accumulation interannual variability.*

We agree with the reviewer in general. An analysis of the interannual variability would be very interesting and insightful. However, the data sets available in this study (annual in-situ point mass balance, decadal geodetic mass balance, repeated snowline observations) only allow us to directly constrain annual mass balances. Seasonal and daily mass balances, calculated by the snowline-constrained model are not tied directly to any observations and day-to-day variability depends on partly uncertain meteorological information. Therefore they are subject to larger unknown uncertainties. Furthermore, the calibration of the two parameters $DDF_{snow}$ and $C_{prec}$ are not strictly independent and thus could lead to important under- or overestimations of the two seasonal components of the modelled surface mass balances. This shortcoming of the method is described in Section "Discussion" and shown in Figure 13b. Unfortunately, at the current stage, we only have a few seasonal in situ measurements for the studied glaciers. A reliable verification of the performance of the snowline approach in terms of the winter and summer surface mass balance is thus not possible. For the above-mentioned reasons we avoid an interpretation of the interannual variability here.

———

*(7) Application of the proposed approach. You mention that the approach you propose can be useful to quantify surface mass balance time series for a number of remote glaciers. Although I mostly agree with this statement, I wonder if the approach can indeed be applicable for summer accumulation glaciers (like in the tropics, or in monsoon regime regions) or for high latitudes glaciers where superimposed ice can be more important than in Kyrgyzstan. You can probably add some sentences on this point in the conclusion.*

We added some sentences on the potential wider applicability of the proposed approach in the Discussion section of our manuscript. However, we are unable to perform a complete assessment of the transferability of our approach to other climate zones in the scope of the present study.

**Page 24, Lines 17-24**
"SMBs inferred from the snowline approach are closely tied to the representativeness of snowline observations. The method might be able to yield reliable SMB estimates for many glaciers in different climatic regimes, for which the transient snowline is an indicator of the surface mass balance. The relationship between the snowline and the SMB can however be importantly challenged when the position of the transient snowline is blurred by fresh snow or superimposed ice. The applicability of the snowline approach presented here can thus be critical when the transient snowline on remote sensing data cannot unambiguously be identified. This is mainly a problem for glaciers with a summer accumulation regime due to frequent fresh snow falls, and glaciers with a high relevance of superimposed ice."

———

Specific comments:

Abstract

*P1, L5-6: the sentence "A combination of 3 independent [] Golubin and No. 354"needs to be reformulated. Indeed, the methods are not combined to reconstruct the surface mass balance. The methods are compared/cross-validated but not combined. For me, you would have a combination if, for example, your modeled surface mass balance time series had been adjusted with the geodetic method.*

Reformulated.

**Page 1, Lines 5-7**
"By cross-validating the results of three independent methods, we reconstructed the mass balance of the three benchmark glaciers, Abramov, Golubin and No. 354 for the past two decades."
————

*P1, L8-9: "satellite optical imagery"instead of "satellite imagery".*

Done.
————

*P1, L12, 13 and in the entire paper: should write "yr-1", instead of "a-1". "yr-1"is most common except for IGS journals.*

Done.
————

*P1, L15: prefer "unmonitored"to "inaccessible". I do not know about an inaccessible glacier on Earth."*

Done.
————

1. Introduction

*P2, L2: should the IPCC reference be quoted Stocker et al. (2013)*

Referencing has been changed.
————

*P2, L18: remove a comma after "e.g.,"*

Done.

————

2. Study site and data

*P4, L12-13: this statement is useless if you do not give any quantification.*

Quantification has been added.

**Page 4, Lines 11-13**
"For Golubin, the geodetic mass loss reported by Bolch (2015) was $-0.28 \pm 0.96\,\mathrm{m\,w.e.\,yr^{-1}}$ from 2000 to 2012, whereas Brun et al. (2017) found a geodetic mass balance of $-0.04 \pm 0.19\,\mathrm{m\,w.e.\,yr^{-1}}$ for the period 2002 to 2013."

————

*P5, L11-12: idem, give a value.*

Quantification has been added.

**Page 5, Lines 11-13**
"Mass loss since the mid-1970s was reported by different studies ranging from about $-0.8$ to $-0.5\,\mathrm{m\,w.e.\,yr^{-1}}$ (Brun et al., 2017; Kronenberg et al., 2016; Pieczonka and Bolch, 2015) ."

————

*P5, L6 and 15: should mention the elevation of the AWS*

Done.

————

*Figure 3: for Glacier No. 354, the Quickbird and Pliades images recorded in 2003 and 2015 respectively have not been used for snowline mapping? If yes, these images should appear in the Figure. If not, why?*

The QuickBird image has not been used for snowline mapping because it dates from summer 2003, and the first year to compute was the year 2004. We clarified in the caption and manuscript text.

**Page 7, Figure 3**
"Image availability and distribution for snowline mapping. Numbers indicate the total available scenes per year and glacier. Prior to 1998, image coverage is sparse for all three glaciers. For Golubin and for Glacier No. 354, the first summer season for which enough snowline observations could be collected was 2000 and 2004, respectively. Snow-covered high-resolution images have not been used to delineate the snowline and are not shown here."

————

*Table 1: should write "snowline"instead of SNL. You should also indicate the sensor in brackets for the high resolution images. If I am correct these are SPOT5 and Pliades for Abramov, ALOS*

*and SPOT7 for Golubin and Quickbird, GeoEye and Pliades for No. 354.*

Done.
————

3. Methods

*See my main comment related to this section (no4), and the one regarding the terminology for the title of this sub-section.*

Please refer to the answer given for the main comment No. 4.
————

*P10, L23: the paper by Huintjes and others has not been finally published. I wonder if papers that stayed in discussion and/or were rejected can be quoted. You can probably remove this reference from the text and the ref list, all the more that it is quoted within a list of 5 references starting by e.g.*

The reference has been removed.
————

*In Table 2 and in the text (e.g. P11, L7): I recommend using Z instead of H for the elevation criteria. H usually stands for thickness.*

Done.
————

4. Uncertainties and model sensitivity

*P15, L3-8: you should provide an illustration for the test that uses average daily temperature and precipitation series. In addition, you have to explain the low sensitivity of your model to the input meteorological data. I assume this is because the parameters are adjusted for each year and for each glacier.*

An illustration has been added and we have extended our explanation on the low sensitivity of the model to the meteorological input data (see Fig. 6)

**Page 15, Lines 27-30**

"These results demonstrate a relatively low sensitivity of our model approach to daily meteorological input data. With the chosen calibration procedure the model parameters $DDF$snow and $C_{\mathrm{prec}}$ are adjusted to best represent the TSL observations for each year and glacier individually. The modelled SMB are thus closely tied to the snowline observations and exhibit a reduced dependence from meteorological input data."
————

*Figure 5b: change JAJ by JJA*

Done.

————

5. Results

*P16, L3-6: you should provide an illustration for the comparison between SCAFs given by the model and the images for all the used images, glacier by glacier.*

We added an illustration comparing the observed and modelled SCAF for each glacier (see Fig. 7).

————

*P16, L7-8: the period 1998 to 2016 stands for Abramov Glacier only. You have to mention the specific periods for each glacier. Golubin starts in 2000 and No. 354 in 2004. In addition "over the two last decades"can be removed from the sentence because the time periods for each glacier will be mentioned.*

Done.

**Page 18, Lines 3-5**
"Annual glacier-wide modelled surface mass balances constrained by snowline observations, calculated for Abramov (1998-2016), for Golubin (2000-2016), and for Glacier No. 354 (2004-2016), located in the Pamir-Alay and the Tien Shan, are predominantly negative (Fig. 8 and Tabel 6)".

————

*P16, L8: you refer to Figure 8, but because figures 6 and 7 have not been quoted yet, this figure should be Figure 6. Anyway, because I suggest adding two more figures, it will probably remain Figure 8, but must appear before the current figures 6 and 7.*

Numbering has been corrected.

————

*Table 5: you must indicate in the table itself (not only the caption) on which period the STD is quantified*

We removed the STD from the table but included a statement directly into the text.

**Page 18, Lines 12-14**
"A lower standard deviation of annual mass balances from 2004 to 2016 is found for Glacier No 354 ($0.19\,\mathrm{m\,w.e.\,yr^{-1}}$) than for Golubin ($0.4\,\mathrm{m\,w.e.\,yr^{-1}}$) and Abramov ($0.29\,\mathrm{m\,w.e.\,yr^{-1}}$), which indicated higher interannual variability for the latter two."

————

*P17, L1: should the first close-to-zero SMB period be extended to 2005?*

We agree and extended the period to 2005.
————

*P17, L1: "Glacier No. 354, situated in a more continental climate regime, []". I am a bit skeptical with this statement! All three glaciers are in a continental regime. This glacier being located in the inner range, it might receive less precipitation than the two others. Is it what you mean? See also my main comment related to interannual variability of the SMB (no6).*

Indeed, we refer to a strong precipitation gradient from West to East for both the Pamir/Pamir-Alay and the Tien Shan and also a different precipitation distribution. Glaciers in the western part of the Tien Shan receive more winter accumulation whereas glaciers located more in the east are subject to considerable summer accumulation. This corresponds to the general synoptic large-scale meteorological conditions over Central Asia, influenced by the main direction of the zonal flow of the air masses from west to east. According to Balashova et al. (1960) and Schiemann et al. (2008) also meridional air mass flow can occur. This occurs either in situations when tropical air masses enter from South and south-west or when north-westerly, northerly and sometimes even north-easterly, cold air masses intrude into Central Asia. We specified the statement but do not want go into more detail in the manuscript.

**Page 18, Lines 14-15**
"Glacier No. 354, receiving lower amounts of total annual precipitation, showed a smaller mass turnover and had a positive balance only in 2009 (Table 6)."
————

*P17, L4-5: this sentence refers to the years 2006 and 2008 mentioned in the previous sentence? If yes, the two sentences might be separated by a semi-colon not a dot.*

Done.
————

*P17, L8-9: same thing here, the two sentences could be separated by a semi-colon not a dot.*

Done.
————

*Table 6: you should mention in the caption that the values differ from Table 5 because the glacier-wide annual surface mass balances are not computed over the same number of days. However, the difference is really high for some years, for example 2014 for Golubin Glacier (more than 0.8 m w.e. different!). You could indicate the number of days differing from the quantification given in Table 5.*

Instead of indicating the number of differing days, we added a table with the exact survey dates, and changed the statement in the caption to underline the difference to Table 5.

**Page 20, Table 7**
"Annual SMB $B_{sfc(meas)}$ for the measurement periods (i.e., exact dates of the surveys Table 2) based on direct

glaciological measurements and on the snowline-constrained model for the three glaciers in $\mathrm{m\,w.e\,yr^{-1}}$."

———————

*Table 7. A dot is missing after e in "m w.e a-1"In addition, regarding Abramov Glacier, why the first period is 2003-2015 and not 2003-2011?*

Due to the limited image quality of the SPOT image from November 2011, we considered the mass balance from 2003 to 2015 to be more robust, and decided to show the result for this period instead.

———————

6. Discussion

*P20, L3: replace "integrating"by "using"*

Done.

———————

*P21, L6: discuss why the difference is opposite for Glacier No. 354*

We discovered an error in the model settings of Glacier No. 354 of the unconstrained model leading to the too positive balance for Glacier No. 354. The error is corrected and we would like to apologize for this mistake. The calculations and figures are now updated (see Fig. 12). Not surprisingly, the mass balance is much more negative for the unconstrained run.

———————

*P22, L27: replace "too positive"by "not negative enough"*

Done.

———————

*P22, L33: change "shows"by "showed"*

Done.

———————

*P23, L18-20: you indicate that the average glacier-wide annual surface mass balance quantified by Brun et al. (2017) shows a stronger mass loss than your study. This is not really exact. The difference is important with your modeling approach, but the estimate by Brun and others and your geodetic estimate are really close.*

We clarified our statement.

**Page 26-27, Lines 30-2**
"The average mass balance for Abramov of $-0.38 \pm 0.10\,\mathrm{m\,w.e.\,yr^{-1}}$ (2002-2014) derived by Brun et al. (2017)

using multi-temporal ASTER DEMs indicates a stronger mass loss than the results obtained with the snowline approach. We note, however, that the start and end dates of their geodetic mass balance assessment represent a mean over a mosaic of different dates, thus hampering direct comparison. In addition, the differences are still within their error bounds. The results by Brun et al. (2017) are, in line with the geodetic mass balance calculated in the present study for the period 2003 to 2015 based on high-resolution satellite images."

————

**Comments of Reviewer #2 (Prof. Dr. M. Pelto)**

*2-8: signification to significance*

Done.
————

*2-15. Be consistent on spelling of Urumchi*

This has been corrected through the entire manuscript. We use Urumqi.
————

*3-3: Change Pelto (2011) toPelto et al. (2013)*

Done.
————

*Figure 3: Conveys important information, this could just as easily be conveyed in a table if there are production advantages to that*

We prefer to illustrate the image availability in a figure rather than in a table. The individual image dates are not very important in this context. However, we would like to sketch the increased image availability with time which, in our opinion, becomes more evident with a figure.
————

*5-14: Is the superimposed ice evidence indicative of persistent or transient existence?*

The superimposed ice zone is rather persistent. We clarified this in the manuscript.

**Page 5, Lines 14-15**
"Evidence of persistent superimposed ice is found and Kronenberg et al. (2016) estimated internal accumulation to be $+0.04\,\mathrm{m\,w.e.\,yr^{-1}}$."
————

*7-7: Later in the paper it is worth simply mentioning the overall retreat observed on the three glaciers and how that fits into the negative balance regime.*

We added some statements on the retreat pattern of the glaciers and brought it into context with the negative balance regime in the Discussion section.

**Page 18, Lines 6-12**
"For Golubin and Glacier No. 354 a slightly more negative annual average balance of $-0.41\pm0.33\,\mathrm{m\,w.e.\,yr^{-1}}$ and $-0.36\pm0.32\,\mathrm{m\,w.e.\,yr^{-1}}$, respectively, was calculated for the same time period (Table 6). Length change

measurements underline the observed negative balance regime of all three glaciers (Hoelzle et al., 2017). A significant glacier retreat was observed for the last century. A first speed-up of frontal retreat occurred in the 1980s and acceleration was observed in the last decade. However, no clear acceleration of mass loss for the three glaciers was identified over the investigated periods. Two phases of close-to-zero SMB could be recognized (2002-2005 and 2009-2011) for all glaciers."

————

*8-18: Given that a couple of the references are related to the study I would add Mernild et al. (2013) Pelto et al. (2013) where this is also discussed and are already references used elsewhere in paper.*

Done.

————

*8-25: Is the contrast with snow to firn weak for the terrestrial camera or just satellite images?*

Clarified.

**Page 8, Lines 29-31**
"Contrast becomes rather weak, especially when the snowline rises above the firn line (Rabatel et al., 2013; Wang et al., 2014) for both satellite and terrestrial camera images."

————

*9-4: The initial position being the GPS location at the time of emplacement?*

The GPS measurements of the glacier front position were repeated every year.

**Page 8, Lines 1-2**
"Annually repeated measurements of the glacier front position using a handheld GPS for all three glaciers were combined with the satellite observations for mapping from 2011 onward."

————

*9-9: The extrapolation indicated is spatial. Given the field seasons occur before the end of the ablation season, is this also a temporal extrapolation model or is this different. Just clarify temporal from spatial extrapolations.*

Clarified in the manuscript. The extrapolation is only spatial. We did not adjust the glaciological mass balance for the start and end date of the hydrological year, but calculated the mass balance derived with the snowline approach to match the dates of the direct measurements for comparison.

**Page 9, Lines 10-15**
"A model-based spatial extrapolation of point measurements to the entire glacier surface after Huss et al. (2009) was used to retrieve glacier-wide SMB for all years with direct measurements. The model is a combined distributed accumulation (Huss et al., 2008) and temperature-index melt model with daily resolution (Hock, 1999) which was

automatically optimized to best represent all collected point data from each seasonal/annual survey. The model is considered as a suitable tool to extrapolate the glaciological point measurements to the glacier surface for the measurement periods."

————

*9-30: Given the issues described what is the vertical accuracy generally achieved? This is I think discussed at 13.24, but is appropriate here too.*

We added the vertical accuracy.

**Page 10, Lines 6-8**
"The vertical accuracy was thus improved, and the mean absolut difference off-glacier was limited to 1.0 m (2003-2015) and 0.6 m (2011-2015) for Abramov, 0.7 m for Glacier No. 354 and 1.6 m for Golubin (see also Section 4)."

————

*10-8: Is the approach what Pelto (2010) and Mernild et al. (2013) utilized which is TSL migration rate * balance gradient? This yields a directly observed ablation rate.*

No, we did not use the same approach as presented in Pelto (2010) and Mernild et al. (2013). These approaches integrate direct field measurements to calculate snow ablation rates which are not consistently available for our study region. Here, we use an iterative modelling approach to calculate snow accumulation and use only TSL as input. Please refer to Section 4.3 that gives details on the model constrained by snowline observations.

————

*11-1: Why is the DDFs for Golubin very close to the maximum?*

Thanks for spotting this. It was an error in the submitted manuscript. The mean DDF is $5.09 \, \mathrm{mm \, day^{-1} \, {}^{\circ}C^{-1}}$ and not $5.49 \, \mathrm{mm \, day^{-1} \, {}^{\circ}C^{-1}}$. We apologize for this mistake.

————

*13-6: Geodetic mass balance calculations do not account for internal accumulation either, unless it is incorporated in the density calculations, which typically does not occur.*

This is a very interesting comment on the problem of geodetic mass balance computations. We are however unable to fully resolve this issue in the present paper. We completely agree with the reviewer that the density assumption to calculate the geodetic mass balance is critical and so far not well understood. In principle, the geodetic mass balance, however, includes all mass changes within a glacier, and not only surface processes. The problem goes back to the density assumptions of converting volume to mass changes.

We are sure that a correction is needed for unifying surface mass balances with geodetic surveys that monitor all mass change components. However, as stated by the reviewer, to correctly account for internal accumulation within the geodetic mass balance, a correct density assumption is required. Yet, this is not straight-forward, and we believe that improving the confidence in

volume-to-mass change density assumptions is not possible within the scope of this study and more process-related studies on this subject are required. We are aware that chosen density assumption is a strong simplification but at the current stage, we are simply not able to reasonably correct the calculated geodetic mass balance for the component of internal accumulation. This is why we have chosen an error ranges for the volume-to-mass conversion which are expected to cover the respective uncertainties.

————

*13-29: Why this choice of 120 kgm-3, and what are implications vs a less conservative choice?*

As mentioned above, the density assumption is very critical to convert volume to mass change. Currently, we have unfortunately not enough knowledge to make more adequate assumptions. Nevertheless, we decided to use a more conservative volume-to-mass conversion and simply doubled the uncertainty range of the density for periods shorter than 3 years.

————

*15-7: "These results demonstrate a relatively low sensitivity of the presented model to daily meteorological input data compared to mean seasonal data...".*

The sentence has been changed according to the comments of reviewer 1.

**Page 15, Lines 27-30**
"These results demonstrate a relatively low sensitivity of our model approach to daily meteorological input data. With the chosen calibration procedure the model parameters $DDF_{\text{snow}}$ and $C_{\text{prec}}$ are adjusted to best represent the TSL observations for each year and glacier individually. The modelled SMB are thus closely tied to the snowline observations and exhibit a reduced dependence from meteorological input data."

————

*16-10: Section 4 was an excellent detailed summary of the approach to determining errors and sensitivity. How did that lead to the error numbers here which are somewhat higher than I expected after seeing the details in Section 4.*

We combined the different error sources for each year by the Root Sum of Squares, assuming independence between the different error components and averaged the annual errors for the considered periods. We clarified this in the manuscript.

**Page 17, Lines 1-3**
"Components 1 to 5 are assumed to be independent of each other and are combined as RSS to represent the total error of the annual SMB $\sigma_{\text{tsl}}$ obtained from the snowline approach. We then averaged the annual error over the different periods to compute overall uncertainty."

————

*18-9: The more negative balance years on Golubin and Abramov is where the TSL method generates more negative results. Could this be a reflection of melt low on glacier outside of ablation season that TSL does a better job of capturing? If this cannot be addressed to advantage than*

*do not.*

As the daily mass balance evolution is a pure product of our modelling approach and per se not actually constrained by the available observations (in contrast to the annual mass balance), we prefer to leave aside interpretations on the seasonal components of the surface mass balance (see also comment to reviewer 1 above).

————

*21-1: Could utilize Shea et al. (2015) as well for support they found almost the same value for the Mount Everest region, different climate setting but still a high altitude monsoon influenced area. Wu et al. (2011) also determine DDFs for Urumqi Glacier that could be referenced*

We integrated the suggested references to underline the choice of our parameters.

**Page 22-23, Lines 6-3**

"We chose $C_{\text{prec}}$ to account for a 20%-measurement error of observed precipitation (Sevruk, 1981), and a combination for $DDF_{\text{ice}}$ ($7.0\,\text{mm day}^{-1}\,{}^\circ\text{C}^{-1}$) and $DDF_{\text{snow}}$ ($5.5\,\text{mm day}^{-1}\,{}^\circ\text{C}^{-1}$) as recommended by Hock (2003) for the former Soviet territory, for all three glaciers. Similar values were used to model glaciers in the Tien Shan and Himalayas (e.g. Shea et al., 2015; Wu et al., 2011; Zhang et al., 2006)."

————

*21-6: Why the large divergence for Abramov Glacier after 2009 between the constrained and unconstrained model?*

2009 was a rather cold and snow-rich year. Especially summer snowfall reduced the melt for most glaciers in the region (Barandun et al., 2015; Kenzhebaev et al., 2017; Kronenberg et al., 2016). A less negative mass balance can be observed for all three glaciers in the results obtained from both models.

The divergence between the constrained and unconstrained model increases after 2011. SMB are much more negative for the unconstrained model from 2011 to 2016 than before. This is most likely due to the change of the meteorological input datasets. In 2011, the new meteorological station was installed, and we replaced the Reanalysis temperature data with measured air temperature. With the use of the snowline data the change of the meteorological data is secondary because of the decreased sensitivity towards the meteorological input. However, when using an unconstrained model, such effects can be quite large as shown in Figure 12.

————

*22-15: The TSL observations also represent a direct point balance observations of considerable value.*

We agree with the reviewer. However, the use of the transient snowline as point mass balance observation is not the focus of this paper and we prefer not to provide more details on this subject here in order to keep our article focussed.

————

*23-9: Consult and refer to Bazhev (1973) who directly measured the internal accumulation in firn on a glacier in the Pamirs and Abramov glacier. Found that almost all meltwater refroze in upper four layers of firn and the amount was in the 0.20 m/a range, this supports the approached used here. Further it is worth mentioning that such a study of internal accumulation should be redone some year as part of the mass balance program. Miller and Pelto (1999) observed a reduction in internal accumulation, on Lemon Creek Glacier, Alaska which if it occurs has impacts on the energy balance.*

For the estimate by Barandun et al. (2015) from where we adopted the estimate of internal accumulation, the study by Bazhev (1973) was used for calibrating the refreezing model. In Barandun et al. (2015), the quantification of refreezing is based on calculating a temperature profile in firn and ice using the heat conduction equation (see, (e.g. Pfeffer et al., 1991)). The refreezing model is calibrated by adjusting firn temperature at the bottom of the profile at model initialization to match repeated firn temperature measurements made in three firn cores (Glazirin et al., 1993). The results were finally compared to findings for Abramov presented by Bazhev (1973).

During recent field visits, we measured firn temperature at two locations in the accumulation zone. No negative temperatures during summer field campaign where found indicating that all energy available for refreezing melt water had been used. A new project (since April 2017) aims at analysing the firn stratigraphy, and at quantifying of refreezing on Abramov Glacier. However, no results are available yet. We thus decided not to add more information on the calculated internal balance but refer to published former work.

———————

*Figure 12: I do not think this is needed, if included redesign as the visual message is not well communicated by the approach used.*

We would like to keep the figure in the manuscript. We are nevertheless very open for suggestions to better communicate the visual message.

———————

*25-12: Worth noting that a SCAF time series such as in Figure 5 is also a critical value for water resource modelling, because of the different DDFs and DDFi values.*

We agree that the method carries a large potential to retrieve mass balance information at higher temporal scale that might be useful for water resource management. However at the moment, the uncertainties related to the daily mass balance series are high and we prefer to refer only to the annual balance components (see also comments above).

———————

[revised manuscript text omitted]

---

## Author Response (AR2)

**Coverletter**

**Multi-decadal mass balance series of three Kyrgyz glaciers inferred from modelling constrained with repeated snowline observations.**

Martina Barandun, Matthias Huss, Ryskul Usubaliev, Erlan Azisov, Etienne Berthier, Andreas Kääb, Tobias Bolch and Martin Hoelzle

**Dear Editor,**

Thank you very much for the editing of our manuscript and your command on the model input data. We added a statement in the abstract and the conclusion to clarify the use of meteorological input data, as suggested. Changes are highlighted in the track-change version and summarized below. However, we would like to emphasize that the insensitivity of the model to meteorological data (and their accuracy) is demonstrated by two independent tests: (1) the perturbation  $(\pm 1^{\circ}C$  for temperature and  $\pm 25\%$  for precipitation) of the meteorological input data, and (2) the use of the long-term climate mean described in Section 4.3 under point 4 of the uncertainty calculation. The latter showed that the model is able to reproduce annual mass balance with satisfying accuracy when using the long-term climate mean instead of the daily meteorological variables. Thus, we are very confident that the approach can be applied on unmeasured glaciers without local meteorological measurements using for example non-adjusted Reanalysis time series. However, we understand that it might not be proven with sufficient credibility in the presented manuscript since we relied on the available measurements for the simulations to guarantee the best estimate of the mass balance series for the three glaciers instead of using Re-analysis datasets only.

We hope to have adequately taken into account your comment in the updated version of the manuscript and would like to thank you once more for your help to improve our study.

Yours sincerely, the authors

**Page 1, Lines 9-12**

"By combining modelling with remotely acquired information on summer snow depletion, it was possible to infer glacier mass changes for unmeasured years. The model is initialized with daily temperature and precipitation data collected at automatic weather stations in the vicinity of the glacier or with adjusted data from climate Reanalysis products."

**Page 26, Lines 22-24**

"We proposed a methodology to derive glacier SMB series for unmeasured glaciers based on mass balance modelling constrained by repeated snowline observations relying either on in situ temperature and precipitation data or climate Reanalysis datasets."